# A stochastic parameterization of ice sheet surface mass balance for the Stochastic Ice-Sheet and Sea-Level System Model (StISSM v1.0)

Lizz Ultee[1,2], Alexander A. Robel[1], and Stefano Castruccio[3]

[1]School of Earth and Atmospheric Sciences, Georgia Institute of Technology, Atlanta, GA, USA
[2]Department of Earth & Climate Sciences, Middlebury College, Middlebury, VT, USA
[3]Department of Applied and Computational Mathematics and Statistics, University of Notre Dame, Notre Dame, IN, USA

**Correspondence:** Lizz Ultee (eultee@middlebury.edu)

**Abstract.** Many scientific and societal questions that draw on ice sheet modelling necessitate sampling a wide range of potential climatic changes and realizations of internal climate variability. For example, coastal planning literature demonstrates a demand for probabilistic sea-level projections with quantified uncertainty. Further, robust attribution of past and future ice sheet change to specific processes or forcings requires a full understanding of the space of possible ice sheet behaviors. The wide sampling required to address such questions is computationally infeasible with sophisticated numerical climate models at the resolution required to accurately force ice sheet models. Stochastic generation of climate forcing of ice sheets offers a complementary alternative. Here, we describe a method to construct a stochastic generator for ice sheet surface mass balance varying in time and space. We demonstrate the method with an application to Greenland Ice Sheet surface mass balance, 1980-2012. We account for spatial correlations among glacier catchments using sparse covariance techniques, and we apply an elevation-dependent downscaling to recover gridded surface mass balance fields suitable for forcing an ice sheet model while including feedback from changing ice sheet surface elevation. The efficiency gained in the stochastic method supports large ensemble simulations of ice sheet change in a new stochastic ice sheet model. We provide open source Python workflows to support use of our stochastic approach for a broad range of applications.

## 1 Introduction

Many decision-making contexts demand probabilistic projections of sea-level rise. For example, urban planners managing coastal risks would like to be able to quantify the probability of certain levels of sea-level rise (Walsh et al., 2004) so that they can apply their own risk tolerance to assess proposed interventions (Kopp et al., 2014; Hinkel et al., 2019). Probabilistic projections can also help illustrate the benefits of climate mitigation actions for policy-makers, quantify coastal adaptation needs, and identify priority areas for further research (Jevrejeva et al., 2019, and references therein). Efforts to generate probabilistic projections of future sea level change have been ongoing for decades (Titus and Narayanan, 1996), but the ice sheet component remains a source of poorly quantified uncertainty (Le Cozannet et al., 2017; Sriver et al., 2018; Jevrejeva et al., 2019).

Generating probabilistic projections of ice sheet contribution to sea level requires running many climate and/or ice sheet model simulations that can explore multiple realizations of an uncertain future. The spectrum of methods available to generate future projections of ice sheet change makes that task difficult. The most computationally efficient methods find an empirical

relationship between some climate variable, often global mean surface temperature, and a variable of interest, such as global mean sea level (Rahmstorf, 2007) or ice sheet melt (Luo and Lin, 2022). Such methods allow wide sampling of future climate scenarios, which is necessary to account for scenario uncertainty. However, they assume that the form of the relationship between the variables will remain the same in the future, which is not assured in a rapidly changing climate with feedbacks among multiple variables. The structural uncertainty in those methods—that is, the uncertainty attributable to poor knowledge of the form of the model itself—is therefore high, and their results are difficult to convert into a probability distribution.

More sophisticated numerical models represent physical processes, such as ice sheet flow, snowfall, and surface melting directly (Goelzer et al., 2020b; Seroussi et al., 2020), explicitly modeling changes over time in the relationship between climate forcing and output variables of interest. Such models include many more parameters and internal variability of processes on a wide range of spatial and temporal scales. A direct representation of physical processes helps to constrain structural uncertainty related to processes and internal variability, but the computational expense of sophisticated models limits the number of future scenarios that can be sampled. Model outputs thus represent discrete points in a wide range of possibilities, providing too little information to estimate the probability distribution of output variables such as future sea level.

The limited sampling available from physical process model outputs has motivated the creation of statistical emulators to explore the probability distribution of ice sheet model output variables (Edwards et al., 2021). To support local sea-level adaptation planning and to guide ice sheet research, it is useful to partition the uncertainty in such probability distributions among various sources — for example, identifying what fraction of the spread comes from uncertainty in the model physics versus what fraction comes from uncertainty in the applicable climate scenario (Jevrejeva et al., 2019; Marzeion et al., 2020). Identifying the fraction of uncertainty attributable to internal climate variability would require large ensemble simulations of ice sheet evolution that sample a representative set of climate forcing fields.

A particular obstacle to large ensemble simulations of future ice sheet evolution is the computational expense of generating surface mass balance forcing. "Surface mass balance" ("SMB") refers to the set of processes through which ice sheets gain and lose mass at the ice sheet interface with the atmosphere. Mass-gain processes include precipitation, vapor deposition, and refreezing of meltwater; mass-loss processes include melting (with subsequent runoff) and sublimation. Due to the complex set of ice-atmosphere interactions that comprise mass balance, ice sheet models are not typically forced directly by global climate model output. Rather, global climate model output must be downscaled to construct an SMB field of high enough spatial resolution and quality, often through use of a specialized mass and energy balance model that accounts for processes at the snow/ice surface and in the snowpack (see e.g. Fettweis et al., 2020, and references therein). Increasing sophistication in the process-based models used to construct ice sheet SMB means a corresponding increase in computational demand for each individual simulation with these models. That added computational expense further limits comprehensive sampling of possible SMB scenarios.

Stochastic methods provide a low-cost alternative to ensembles with multiple realizations of sophisticated process models (Sacks et al., 1989). A stochastic generator can produce a large ensemble sample of SMB comprised of many realizations that are statistically consistent with a small set of process model outputs. Previous studies have applied stochastic methods to analyze ice sheet mass balance observations with the primary aim of testing whether a trend emerges from the range of

natural variability. For example, Wouters et al. (2013) represented SMB simulated by RACMO2 as an order-$p$ autoregressive process to estimate the uncertainty on mass balance trends for the Greenland and Antarctic Ice Sheets. More recent studies tested multiple types of stochastic models to characterize the variability in Antarctic SMB (Williams et al., 2014; King and Watson, 2020) and thereby test the presence of significant, detectable trends in SMB observations. Here, we have a different aim: to construct a statistical generator of SMB to force an ice sheet model. The SMB product we wish to generate should include interannual variability at the catchment scale, temporal trends, seasonality, and spatial variation down to the scale of an ice sheet model mesh. We approximate the output of one process-based SMB model as a realization of a stochastic process. The statistical generator that produces a realization best fit to a given process-based model output can then be used to generate hundreds of other realizations, sampling the range of internal variability for future SMB consistent with the same model, at much reduced computational expense. Those generated samples support large ensemble simulations of ice sheet change, including simplified feedback of ice sheet geometry on SMB (see example application in Verjans et al., 2022). To best support the broader glaciological community, we base our method entirely on open-source software packages and provide our own open-source code where necessary.

Below, we present the data sources that informed our construction of a stochastic surface mass balance generator (§2). We then describe our choice of temporal model type and how we selected the best-fit model for each catchment of the Greenland Ice Sheet (§3.1 - 3.2) . Section 3.3 describes how we accounted for large-scale covariance in SMB across the ice sheet. We demonstrate the generation of forward-projected SMB time series (§3.4) and how to downscale those time series to ice sheet model grid scale (§3.6). Finally, we contextualise our work with previous studies and highlight its potential applications (§4).

## 2    Data

We construct the stochastic SMB model based on SMB fields output from high-resolution regional climate models with domains encompassing the Greenland Ice Sheet. Here, we focus on output from 7 models that participated in the Greenland SMB Model Intercomparison Project ('GrSMBMIP'; Fettweis et al., 2020) to determine whether stochastic generator type/order is dependent on the choice of process model. The subset of GrSMBMIP models we analyse comprises those whose developer team gave us permission to use their archived data for this purpose: ANICE (Berends et al., 2018), CESM (van Kampenhout et al., 2020), dEBM (Krebs-Kanzow et al., 2021), HIRHAM (Langen et al., 2017), NHM-SMAP (Niwano et al., 2018), RACMO (Noël et al., 2018), and SNOWMODEL (Liston and Elder, 2006). This selection includes exemplars of simpler energy-balance models as well as more sophisticated regional climate models (Fettweis et al., 2020), and these models have been extensively validated against observations over recent decades. The GrSMBMIP regional models are all forced at their boundaries by ERA-Interim reanalysis data and have been processed onto a common 1km square grid, with a common ice extent mask applied.

We aggregate each SMB model output field for each outlet glacier catchment at an annual time scale. To achieve that, we overlay each field with the catchment outlines (Figure 1) provided by Mouginot and Rignot (2019) and sum the grid cells that fall within each catchment area, dividing by the total area of the catchment to arrive at catchment mean SMB for each month,

catchment and model from 1980 to 2012. We then sum to annual time scales so that the subsequent analysis produces statistical models of inter-annual variability. SMB variability at the inter-monthly time scale is dominated by the seasonal cycle, which is added back to generated SMB time series through downscaling (Section 3.6).

## 3 Model description

### 3.1 Temporal model for catchment-averaged annual SMB

We fit a generative statistical model for catchment-averaged SMB using an approach adapted from the work of Hu and Castruccio (2021) on other climate fields. We define the $n$ dimensional vector $\mathbf{M}(t)$ to be the catchment-averaged SMB in each of $n$ catchments at time $t$, and we assume that it can be described by an additive model with a temporal variability vector $\boldsymbol{\mu}(t)$ and a noise term vector $\boldsymbol{\epsilon}(t)$ of the form:

$$\mathbf{M}(t) = \boldsymbol{\mu}(t) + \boldsymbol{\epsilon}(t), \tag{1a}$$

$$\boldsymbol{\mu}(t) = \boldsymbol{\beta}_0 + \boldsymbol{\beta}_1 f(t) + \sum_{i=1}^{p} \boldsymbol{\Phi}_i \cdot \mathbf{M}(t-i), \tag{1b}$$

$$\boldsymbol{\epsilon}(t) \sim \mathcal{N}(0, \boldsymbol{\Sigma}). \tag{1c}$$

In the example case we present here, the vectors $\mathbf{M}, \boldsymbol{\mu}, \boldsymbol{\varepsilon}$ have one entry for each of the $n = 260$ catchments at time $t$, and we evaluate each at a total of $m = 30$ time steps.

The temporal trend $\boldsymbol{\mu}(t)$ includes historical mean SMB for each catchment $\boldsymbol{\beta}_0$, and the forcing variable $f(t)$ with a linear coefficient $\boldsymbol{\beta}_1$. The forcing variable, $f(t)$, can be an external process which causes slow changes in SMB, such as atmospheric temperature ($f(t) = T_A(t)$) or simply a prescribed dependence on time (e.g., $f(t) = t$). Finally, Equation 1b includes autoregressive terms up to order $p$ contained in the diagonal matrices $\boldsymbol{\Phi}_i, i = 1, \ldots, p$. The temporal trend as written would thus approximate an autoregressive process of order $p$, AR($p$). Section 3.2 discusses how we identified AR($p$) as the best type of temporal model for this application. At this stage, fitting temporal models to annually-aggregated time series, we exclude seasonal terms from the temporal trend $\boldsymbol{\mu}(t)$; seasonality is incorporated deterministically during the downscaling process described in Section 3.6. All stochasticity in this generation technique enters through inter-annual variability.

The noise term $\boldsymbol{\epsilon}(t)$ is assumed to be independent, identically distributed in time and from an $n$ dimensional normal distribution with mean of zero and covariance matrix $\boldsymbol{\Sigma}$. As we describe in Section 3.3, spatial correlations between catchments are captured in $\boldsymbol{\epsilon}(t)$.

### 3.2 Selecting candidate model type and order

We tested several model types in search of the most appropriate way to represent interannual SMB variability in Equation 1b. Three criteria inform our selection of candidate stochastic model types for temporal SMB variability. First, we would like our

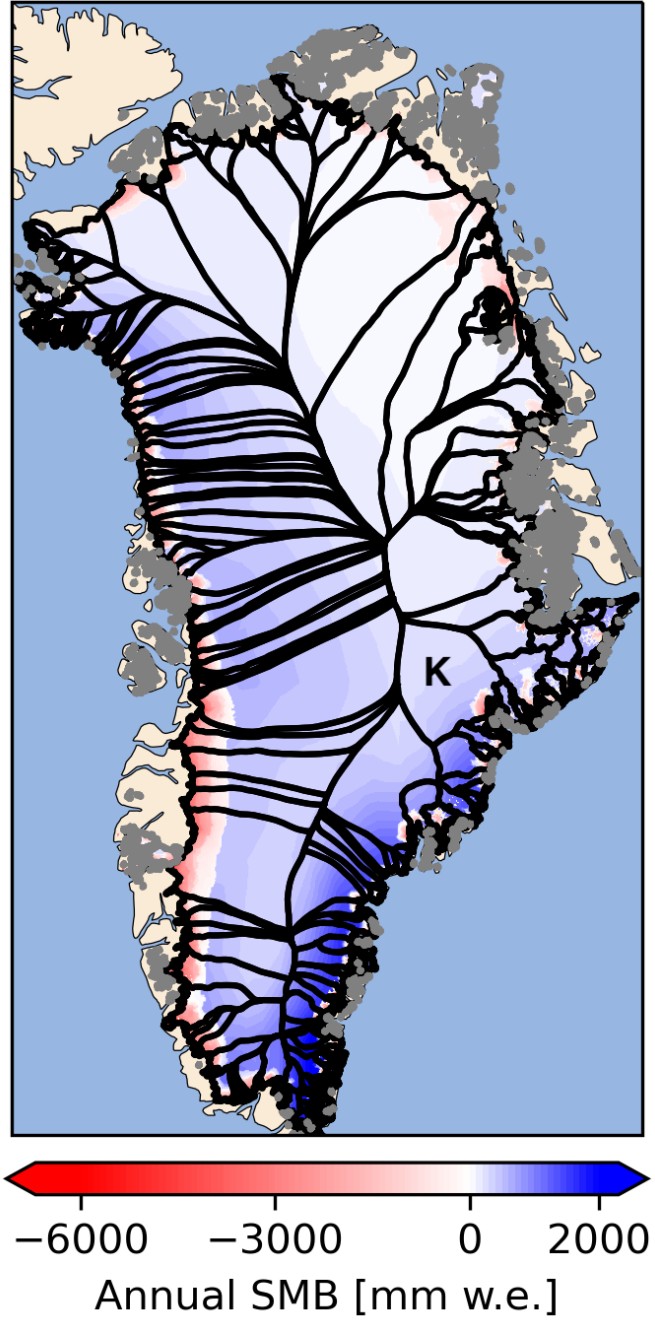

**Figure 1.** Catchments from Mouginot and Rignot (2019) used to aggregate ice-sheet-wide mass balance. Grey outlines indicate catchments that are not simply connected, for example several small glaciers that do not intersect but were grouped together for the Mouginot analysis. Color contour shading illustrates annual mass balance for an example year (2010), taken from the dEBM contribution to SMBMIP (Fettweis et al., 2020; Krebs-Kanzow et al., 2021). The 'K' marker indicates the Kangerlussuaq Glacier catchment, for which example time series and fitting procedure are shown in the following figures.

candidate temporal models to capture the time scales of variability apparent in the data, based on standard statistical methods that are likely to be familiar to glaciologists. Second, we would like our methods to build on existing open-source software, such that other researchers can test and apply our work. For that reason, we prioritize models with existing fitting routines in Python or R. Finally, we would like to be able to compare our findings to those of King and Watson (2020) for the Antarctic Ice Sheet, so we prioritize temporal model families that those authors also tested.

These criteria guide our investigation of three common types of temporal models. All temporal models we test belong to the autoregressive-fractionally integrated moving average (ARFIMA) family of models. The first type, order-$p$ autoregressive "AR($p$)" models are the simplest of the ARFIMA family. They assume that the value of SMB at time $t$ depends linearly on values of SMB at times $(t-1, t-2, ..., t-p)$. An AR(0) model is equivalent to a white noise model scaled to the data. The second type, ARIMA models of order $(p, d, q)$, include order-$p$ autoregressive terms applied to a series that has been differenced $d$ times to reach stationarity, as well as dependence on a weighted moving average of the past $q$ residual noise terms. Finally, general ARFIMA models are similar to ARIMA models but allow non-integer values for $d$, accounting for 'long memory' in the time series. To avoid confusion, we henceforth use "ARFIMA" to refer only to ARFIMA models that do include non-integer differencing $d$, and we refer to the special cases ARIMA and AR($p$) by their own names. King and Watson (2020) tested AR($p$) and ARIMA models; they also tested Generalized Gauss-Markov models, for which we were unable to find an open-source fitting routine, but which are very similar to ARFIMA models of order $(p, d, 0)$.

For each catchment, we estimate $\boldsymbol{\beta}_0$ as the 1980-2012 mean and remove it from the series. We then use Conditional Maximum Likelihood (Ordinary Least Squares) to optimize values of $\boldsymbol{\beta}_1$ and the remaining parameters of Equation 1b associated with each candidate model type (AR, ARIMA, ARFIMA) over a range of orders $(p, d, q)$. We perform the model fitting with built-in functions from the Python package statsmodels v0.12.2 (Seabold and Perktold, 2010): `statsmodels.tsa.ar_model.AutoReg` and `statsmodels.tsa.arima.model.ARIMA`. In each case, we assume a linear dependence on time, $\boldsymbol{\beta}_1 f(t) = \boldsymbol{\beta}_1 t$ in Equation 1b. Statsmodels does not include a built-in function to fit ARFIMA models, so we apply fractional differencing following Kuttruf (2019) and subsequently test ARIMA($p, 0, q$) with the built-in function.

We analyse the Bayesian Information Criterion (BIC) as returned by the statsmodels built-in function for the temporal models fit to each catchment series. The BIC is given by:

$$\text{BIC} = -2\ell + \ln(T)(1 + df), \tag{2}$$

where $\ell$ is the log-likelihood function of the given temporal model on the data, $T$ is the number of observations, and $df$ is the number of degrees of freedom in the generator. Minimizing the BIC balances a maximization of log-likelihood $\ell$ — the probability that a stochastic generator of this type could have produced the data series from the process model being fit — with a penalty for excess parameters (overfitting). We select the temporal model with lowest BIC for each catchment, for each SMB process model. We analyze the preferred temporal model types across all catchment-model pairs to identify the most suitable class of temporal models. We chose to select for minimum BIC to encourage computationally cheap models with fewer parameters (as in King and Watson, 2020); we note that statsmodels also returns other common metrics of model fit such as the Akaike Information Criterion, which could be selected by users with other priorities.

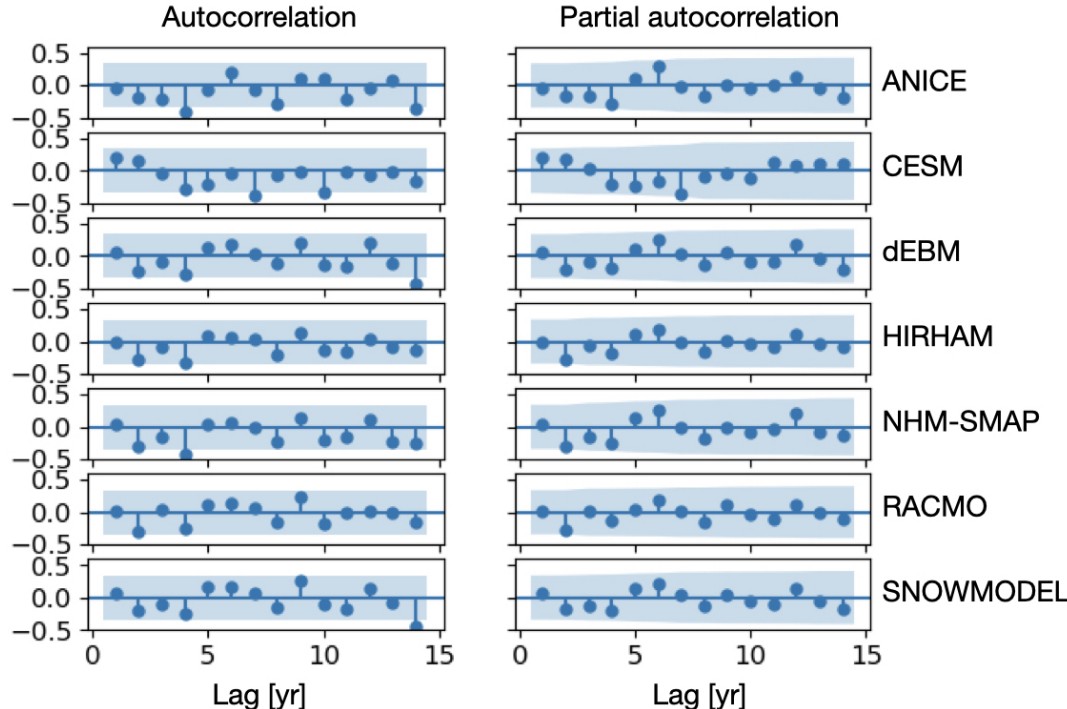

**Figure 2.** Autocorrelation (left) and partial autocorrelation (right) functions for the annual SMB computed by seven GrSMBMIP process models for Kangerlussuaq Glacier, 1980-2012. Lollipop markers indicate ACF or PACF values; the blue shaded region indicates the 95% confidence interval around zero. Values significantly different from zero therefore appear as blue points outside of the shaded region.

To decide the range of orders $(p, d, q)$ to test in our model fitting, we use the autocorrelation and partial autocorrelation functions ("ACF" and "PACF" respectively) to target the relevant time scales of variability. In a purely autoregressive process, the number of values significantly different from 0 before the first non-significant value in the PACF would indicate the AR order $p$. In a purely moving-average process, the number of values significantly different from 0 before the first non-significant value in the ACF would indicate the MA order $q$. These metrics cannot be used to determine the order $(p, d, q)$ of a more general
ARFIMA process, but we use them as qualitative indicators of an appropriate range for testing. The ACF and PACF values differ per process model and per catchment; an example for the Kangerlussuaq Glacier annual SMB is shown in Figure 2. In that example, significant autocorrelation is apparent at a lag time of 4 years for several process models, though with several previous values not significantly different from zero; the partial autocorrelation is not significant for any lag shown. Ice core-derived ACF and PACF show significant values at time scales of up to five years, tapering to values not significantly different from 0 at longer time scales (Figure B1). The combination of evidence from ice cores and from process-model ACF and PACF
in multiple catchments suggests several lags ≤5 years with significant ACF or PACF. We therefore choose to test values of $p$ and $q$ from 0 to 5. We determine the order of differencing required to reach stationarity, $d$, using Augmented Dickey-Fuller and KPSS tests of stationarity on each catchment time series. Both tests agreed that the de-meaned catchment average time series

were stationary, so $d = 0$ should be appropriate, but for completeness we also tested $d = 0.5$ and $d = 1$. Among the range of values $(p, d, q)$ tested, we select the best fit model as the one with the lowest BIC. We note that comparing the BIC of model fits among temporal models of different orders requires a consistent base dataset and fitting method (for example, the same software package and optimization scheme for all models). We computed the BIC using statsmodels built-in functions, setting the optional argument `hold_back` $= \max(p, d, q)$ to ensure that lower-order models were fit to training data series of the same length as higher-order models.

Figure 3b shows example best fits for four model types and their BIC (see legend). The best fit to the NHM-SMAP example data in Figure 3 is white noise with a trend (AR(0)). Both AR($p$) models shown have lower BIC than the more complicated ARIMA and ARFIMA models. In this example, the best-fit AR(0) and ARFIMA series capture a trend with little other temporal information, while the AR(5) and ARIMA(1,0,1) series capture larger temporal variability with the expense of added parameters. We note that the models capturing only a trend in Figure 3b will still generate stochastic series with temporal variability; the distinction is that almost all of the temporal variability in the final generator will come from the spatial noise generation process described in the next section.

In every catchment and SMB model tested (1820 catchment-model pair time series tested), AR models were the most suitable. There were no basin-model pairs where ARIMA or ARFIMA fits were preferred to AR($p$) fits. Further, white noise with a trend (AR(0)) was preferred to any higher-order statistical fit for catchment-aggregated SMB in most basins. Each process model had some basins where higher-order AR($p$) models were preferred (Figure A1).

The example we present below allows the order of the autoregressive model fit to vary by basin. Users may decide to keep that flexibility, which adds some complication in storing the model parameters, or they may opt for the simplest AR(0) fit for every basin and allow residual variability to be captured in the spatial noise generation (§3.3) and downscaling (§3.6).

### 3.3 Estimating SMB covariance between catchments

Thus far, we have described a method for fitting and generating time-varying SMB for individual catchments with no correlation beyond the catchment scale. However, SMB over the Greenland Ice Sheet may vary coherently at spatial scales beyond those of single outlet glacier catchments due to large-scale processes in atmospheric circulation (Lenaerts et al., 2019). Motivated by this physical intuition, we introduce spatially-informed noise generation.

Following Hu and Castruccio (2021), we construct a matrix of variance $\boldsymbol{\Sigma}$ for catchment-level noise terms (Equation 1c above) as:

$$\boldsymbol{\Sigma} = \mathbf{DCD}, \tag{3}$$

where $\mathbf{D}$ is the diagonal matrix of per-catchment standard deviations and $\mathbf{C}$ is the spatial correlation matrix among all catchments. Note that this formulation assumes a catchment-specific variance $\mathbf{D}$, so the SMB is assumed to vary differently within each catchment, implicitly accounting for different catchment sizes. The spatial model $\mathbf{C}$ is defined on the inter-catchment correlation, which we assume not to depend on the catchment size.

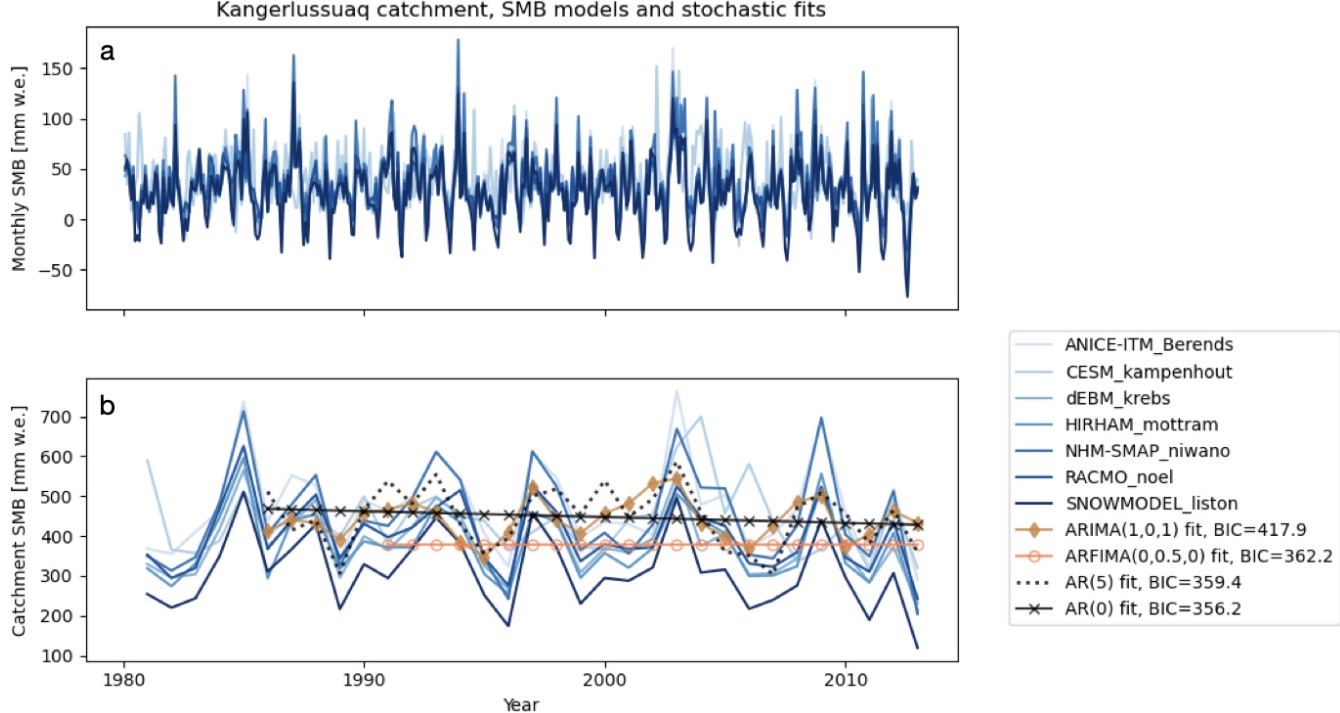

**Figure 3.** Catchment-mean SMB from seven Greenland-wide models. Panel (a): time series at monthly scale, as originally presented in the model output data. Panel (b): time series summed to annual scale, with series from example best-fit stochastic generators overlaid. The Bayesian Information Criterion for each model's fit to an example process model (NHM-SMAP) is shown in the figure legend. Lower BIC values indicate more preferred models.

The spatial correlation pattern may differ for different SMB process models, so we construct the matrix of variance for each SMB process model separately. We calculate the empirical correlation matrix $\hat{\mathbf{C}}$, which is an approximation of $\mathbf{C}$, from the residuals of per-catchment best-fit temporal models described in Section 3.2. We save each residual (length $n = 28$, with 5 years held back from the 33-year training set to accommodate consistent fitting of AR orders up to $p = 5$) as a row in a

260x28 matrix $\mathbf{R}$, one row for each catchment. The empirical correlation matrix $\hat{\mathbf{C}}$ is then the 260x260 matrix of correlation coefficients of the residuals, which we compute using `numpy.corrcoef(R)`. The empirical correlation matrix computed from ANICE-ITM output is shown in Figure 4a.

Because the number of catchments we seek to simulate ($m = 260$ for Greenland) is considerably larger than the number of data points used to train individual statistical models (33 years of catchment-aggregated SMB for each catchment), $\hat{\mathbf{C}}$ is

singular. Therefore, we must enforce a sparsity condition to reduce the influence of spurious information. We estimate a sparse correlation matrix $\mathbf{\Gamma}$ using the graphical lasso algorithm described in Friedman et al. (2007).

We apply the `GraphicalLassoCV` function from the Python package *scikit-learn* v0.24.2 (Pedregosa et al., 2011), which estimates a sparse correlation matrix $\mathbf{\Gamma}$ with the following formulation:

$$\mathbf{\Gamma} = \text{argmin}_K \left( \text{tr } \hat{\mathbf{C}}\mathbf{K} - \log \det \mathbf{K} + \alpha \|\mathbf{K}\|_1 \right), \tag{4}$$

where $\mathbf{K}$ is the inverse correlation matrix and $\alpha$ is a positive regularization parameter. Higher values of $\alpha$ lead to sparser resulting matrices $\mathbf{\Gamma}$. In our implementation, we allow `GraphicalLassoCV` to select the best value of $\alpha$ through cross-validation. Figure 4b shows the sparse correlation matrix resulting from applying this method to ANICE-ITM output.

Each row in the sparse correlation matrix $\mathbf{\Gamma}$ represents the correlation of a given catchment with each other catchment. Figure 4c translates the information in the first row of $\mathbf{\Gamma}$ to a map of Greenland. The first row represents catchment 0 in the Mouginot and Rignot (2019) dataset, Umiammakku Isbræ. Umiammakku has the strongest correlation with itself (dark red shading), moderate positive correlation (lighter red shading) with surrounding catchments and a few more distant catchments, and zero or slight negative correlation (light blue shading) with other catchments in Greenland. We note that these correlations are inferred from the process model data — ANICE-ITM output, in Figure 4 — rather than imposed by physical intuition. As such, the precise structure of the spatial correlation matrix will depend on how the data are aggregated. We would expect slightly different spatial correlations if they were computed with monthly data, or using different catchment outlines. Users must also remember that the spatial correlations shown in Figure 4 are computed on the residuals of temporal model fits, not on the SMB series themselves.

## 3.4 Forward modelling

Finally, we generate a set of realizations of the forward stochastic generator. Each realization is the sum of an autoregressive component and a draw $\boldsymbol{\epsilon}(t)$ from the normal distribution with spatial covariance, as described in Equation 1a and Section 3.3. We find the Cholesky decomposition $\mathbf{\Gamma} = \mathbf{L}\mathbf{L}^{\mathbf{T}}$ of the sparse correlation matrix, and use the lower-triangular component to generate spatially informed noise. The draw $\epsilon_k(t)$ for the $k^{\text{th}}$ catchment is found by matrix multiplication:

$$\epsilon_k(t) = \mathbf{D}\mathbf{L}\,\mathbf{N_j}\,\hat{k} \tag{5}$$

where $\mathbf{N_j}$ is a random normal matrix of shape $(m, Y)$ for $m$ the number of catchments, $Y$ the number of years in the desired time series, and $\hat{k}$ selects the $k^{\text{th}}$ row of the matrix.

We generate realizations of catchment-mean SMB for an example catchment, Kangerlussuaq Glacier. Each realization is a single time series of catchment-mean SMB with variability described by the stochastic generator. Figure 5 shows 10 realizations of Kangerlussuaq SMB from 1980-2050, with process model training data overlaid in black for 1980-2012. By inspection, the stochastic realizations (blue lines on Figure 5) have variability of similar amplitude and time scale to the process model series. The 10 realizations, generated in a few seconds on a laptop, fill the expected range of uncertainty in annual SMB. We interpret that these stochastic realizations are an efficiently generated forcing for ensemble simulations of ice sheet change given SMB subject to internal variability.

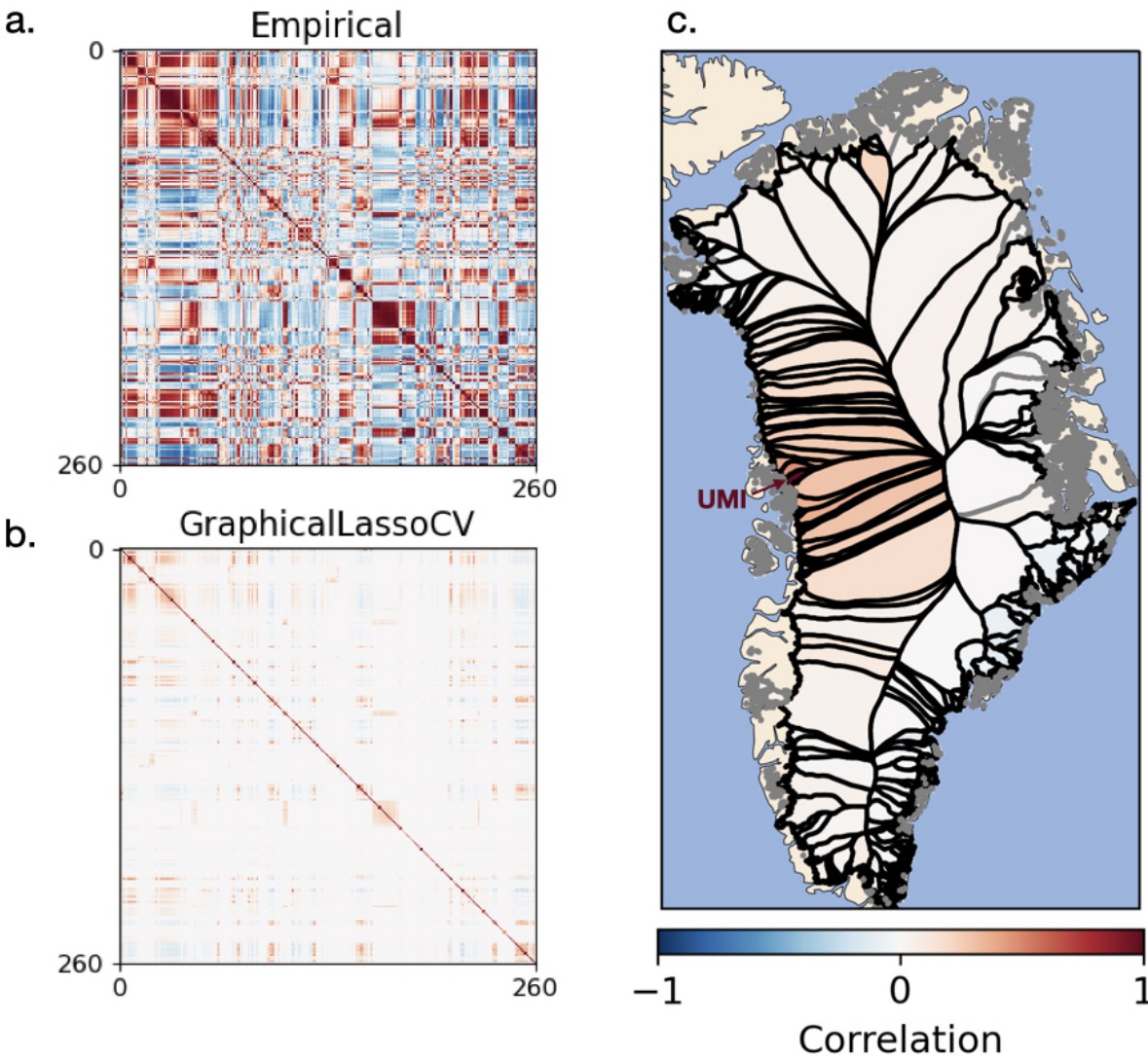

**Figure 4.** Illustration of inter-catchment covariance in ANICE-ITM output data. Panel a: the empirical correlation matrix $\hat{C}$, computed as described in §3.3. Panel b: the sparse covariance matrix that results from applying `GraphicalLassoCV` to $\hat{C}$. Panel c: the first row of the sparse covariance matrix (line 0 in panel b) translated to map view. Catchment 0 is Umiammakku Isbræ, indicated on the map with "UMI" and an arrow to its terminus. All panels share the colorbar shown below panel c.

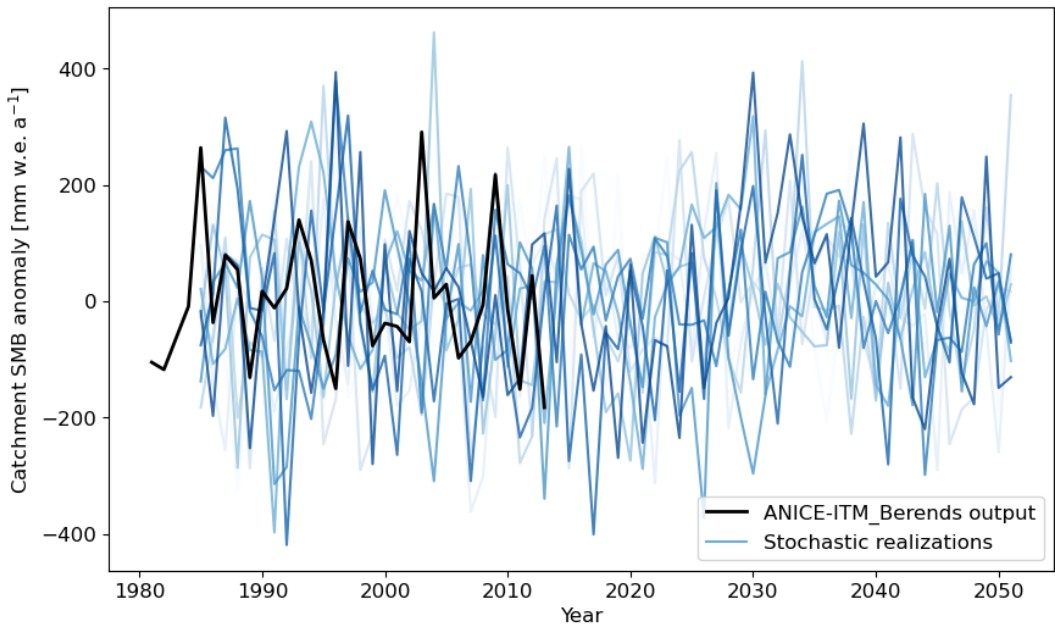

**Figure 5.** Forward simulation of Kangerlussuaq catchment SMB to 2050, with 1980-2010 mean removed, generated using an AR(4) model with spatially-informed noise. The black line shows the results of the process model ANICE-ITM during the period simulated for GrSMB-MIP. Blue lines are single realizations of the stochastic generator.

## 3.5 Non-stationarity

The GrSMBMIP process model historical output we use as our example application did not exhibit non-stationarity, according to the KPSS and Augmented Dickey-Fuller tests applied to each output series (§3.2). We therefore fit stochastic generators that were stationary by construction, assuming the underlying distribution of the data did not change over the period of simulation. We generated stochastic forward simulations as shown in Figure 5 to illustrate the possibility of generating time series with consistent variability outside of the training period. Those simulations fit a linear trend to the training data and assumed that the trend and amplitude of variability remained constant into the future. For scientific applications that study periods of varying climate — for example, glacial-interglacial periods, or century-scale climate projections with anthropogenic forcing — it is expected that SMB time series would not be well fit by stationary models (Weirauch et al., 2008; Bintanja et al., 2020).

To fit a stochastic generator to time series with statistics (mean, trend, variance) varying over time, the user could subdivide the training data series to periods with stationary trend and variance. Piecewise linear trends could be computed on the sub-series, and each series normalized by its variance to create a "z-score" time series. The stochastic temporal model could then be fit to the z-score time series as described above. The output of the stochastic generator would then be re-scaled by the variance in each period to produce ensembles of non-stationary SMB series. The best choice of break points to sub-set the training data will depend on the user's priorities and the time period being investigated; we do not pursue z-score rescaling any further in

this example. For more formal discussion of bias correction in the case of climate data whose distribution changes in time, we refer the interested reader to applied statistics literature e.g. Zhang et al. (2021) and Poppick et al. (2016).

A user generating realizations of SMB at a particular location, or aggregated over some area, could use the method described up to this point. For example, this method could generate realizations of aggregated SMB to support detection of departures from background variability, as in Wouters et al. (2013). The next section describes how to downscale SMB from the catchment annual-average to spatially extensive SMB fields at sub-annual time scales.

## 3.6    Elevation downscaling

To force an ice sheet model, we require a two-dimensional SMB field on the mesh of the model, rather than catchment-aggregated time series. We now apply a spatial and temporal downscaling approach to produce gridded SMB from the stochastically generated series at sub-annual time steps. The downscaling assumes that within each glacier catchment and for a given time of the seasonal cycle, the SMB variation within a catchment can be described by a piecewise linear function with respect to elevation. This downscaling recognizes that, particularly in our Greenland example, there is a strong seasonal cycle in SMB

and that the spatial variations of SMB within a glacier catchment are mostly a function of elevation. As shown in Figure 6, these assumptions are generally quite good for Greenland SMB, and they are reflected in other statistical downscaling approaches that have been previously applied in deterministic frameworks (Hanna et al., 2011; Wilton et al., 2017; Sellevold et al., 2019; Goelzer et al., 2020a). Further, the method generates fields with realistic spatiotemporal variability and elevation dependence, which can be embedded within an ice sheet model (e.g. Verjans et al., 2022) to capture the known feedback between ice sheet

surface elevation change and SMB change (Edwards et al., 2014; Lenaerts et al., 2019).

For each point $\mathbf{p}$ in a given catchment, we need the surface elevation $z(\mathbf{p})$ used to force the physical SMB model underlying our stochastic generator, and the local SMB spatial anomaly

$$\Lambda(\mathbf{p},t) = A(\mathbf{p},t) - \overline{A}(t), \qquad (6)$$

where $A(\mathbf{p},t)$ is the process-model SMB at point $\mathbf{p}$ and time $t$ and $\overline{A}(t)$ is the catchment mean SMB computed from the same

process model at time $t$. We group all local elevation-anomaly pairs by month—for example, all January values together, all June values together—and fit a piecewise linear mass balance gradient for each month $\tau$:

$$\Lambda_\tau(z(\mathbf{p})) = \begin{cases} c_0 + c_1 z(\mathbf{p}) & 0 < z(\mathbf{p}) \leq z_1 \\ c_0 + c_1 z_1 + c_2 z(\mathbf{p}) & z_1 < z(\mathbf{p}) \leq z_2 \\ c_0 + c_1 z_1 + c_2 z_2 + c_3 z(\mathbf{p}) & z_2 < z(\mathbf{p}) \leq z_3, \end{cases} \qquad (7)$$

where $c_0$ is the minimum SMB and the segment slopes $(c_1, c_2, c_3)$ and break points $(z_1, z_2)$ are free parameters optimized by BIC and AIC. In each catchment we thus have twelve functions $\Lambda_\tau$, one for each month. The monthly mass balance gradients

$\Lambda_\tau$ reintroduce seasonal variation. When taken as a function of ice sheet surface elevation $z^*$ that could be evolving in time, they also allow feedback between surface mass balance and dynamically evolving ice sheet geometry.

Example fits for the Kangerlussuaq Glacier catchment, computed from ANICE-ITM output covering 1980-1985, are shown in Figure 6 (and the same example is shown computed from RACMO data in Figure A3). The left panels show spatial anomaly in map view, with the terminus of the glacier to the southeast; the right panels show local SMB departure from the catchment mean as a function of ice surface elevation. The spatial pattern in the example data show strong departures from the catchment mean throughout the lowest portion of the glacier. January SMB in the lower reaches tends to exceed the catchment mean (blue shading); July SMB in the same area tends to be much below the mean (dark red points). Higher elevations show less pronounced departures from the catchment mean (lighter shading).

Finally, we produce time series of monthly local mass balance $a$ for each grid point $\mathbf{p}$ of the $k^{\text{th}}$ catchment:

$$a(\mathbf{p},t) = \mathbf{M}(t) \cdot \hat{k} + \Lambda_\tau(z^*(\mathbf{p},t)), \tag{8}$$

where $t$ is the time in months since the start of the series, $\mathbf{M}(t)$ is the annual catchment-mean SMB generated by the stochastic generator, $\Lambda_\tau$ is the local SMB spatial anomaly for month $\tau$ as defined above, and $z^*(\mathbf{p},t)$ is the local surface elevation at time $t$. The same principle could be adapted for training data provided at even finer temporal resolution, though a large training data set may be needed to capture the relevant variability in sub-monthly SMB.

To illustrate the method, we applied the elevation-based downscaling to estimate local SMB series at two different points, distributed across elevation, in the Kangerlussuaq Glacier catchment. Figure 7b shows those time series. Blue lines are the stochastically generated SMB, downscaled to a single point in space; black lines are the process model output at the grid cell nearest to the selected point. The point represented in the bottom panel is near the terminus and shows large-amplitude seasonal and inter-annual variations in both the process model and the downscaled stochastic realizations. The stochastic realizations track closely with the process model series, while also including inter-annual variability in winter and summer SMB that differs between realizations. The point in the top panel is in the accumulation area. For that point, the range among the stochastic realizations is wider than the apparent variability in the process model series. The seasonal cycle has approximately correct amplitude. We interpret that the variability in the catchment-averaged SMB is dominated by large-amplitude variation near the terminus (Figures 6 and A3), which is then reflected in the stochastic generator fit to the process model series. We further discuss this overestimate of accumulation zone inter-annual variability in the next section.

## 4 Discussion

Simulating ice sheet evolution in a numerical model generally requires a two-dimensional SMB field that may vary in time. Here, we have laid the foundation for efficiently generating many realizations of a time-varying SMB field with stochastic methods. Figure 7 demonstrates that our method can produce realistic SMB time series across an outlet glacier catchment. To produce a two-dimensional field, a user would apply the downscaling method described in Section 3.6 to every grid point in the catchment. The piecewise linear mass balance gradients shown in Figure 6 (insets) are provided to the user as mathematical functions, so the downscaling can be applied on whatever mesh the user provides. This simplicity also allows this method to be incorporated directly into an ice sheet model so that feedback of changing ice sheet geometry on SMB is included, in addition

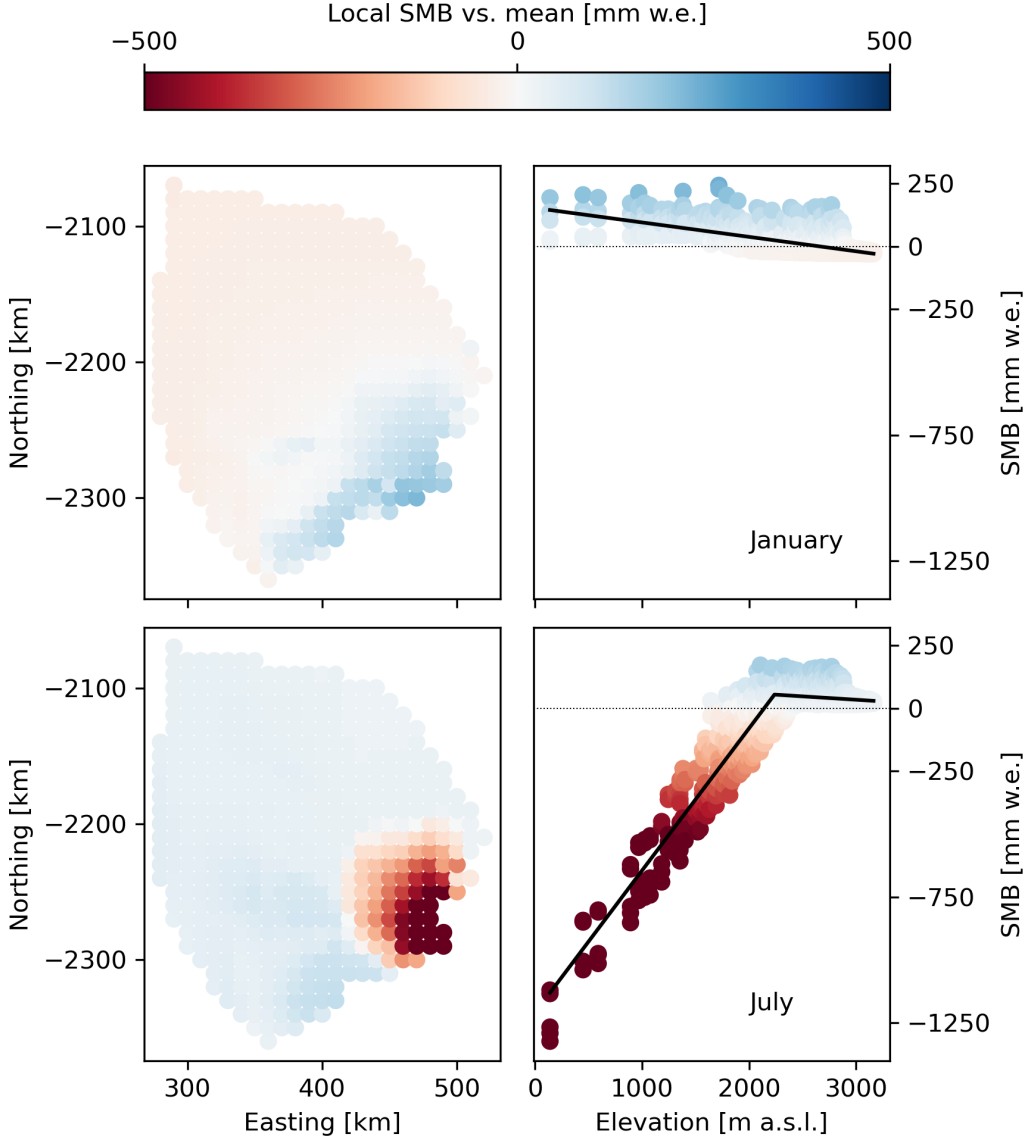

**Figure 6.** SMB downscaling in time and space, shown for two example months (rows): January and July. Left panels: SMB spatial anomaly (difference from catchment mean) for each point within the Kangerlussuaq Glacier catchment, based on the ANICE-ITM model contribution to GrSMBMIP. Right panels: SMB lapse rate with elevation for each month, deduced from the anomaly fields shown. Colored data points represent individual pixel values, 1980-1985. Colormap is consistent across panels.

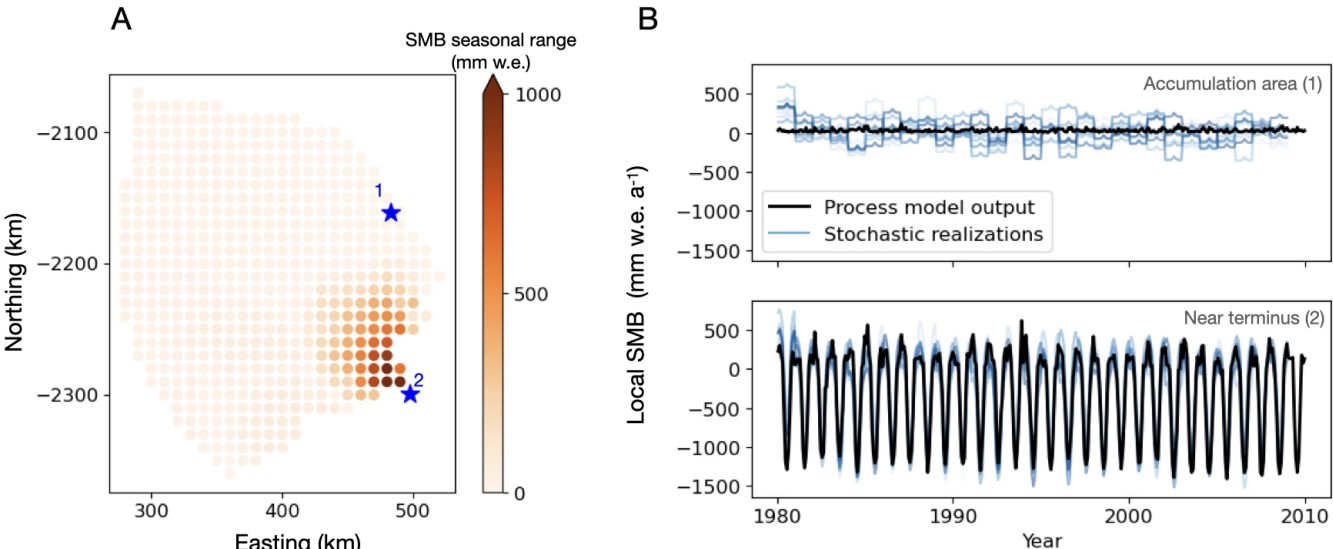

**Figure 7.** (A) Location of example points (blue stars and numbers) in the Kangerlussuaq Glacier catchment. Circle markers show SMBMIP grid points, colored by the seasonal range in mass balance at that location, computed as local mass balance in December minus local mass balance in July. (B) SMB time series scaled from catchment mean down to local (single grid point) values. The upper series are scaled to Point 1, in the accumulation area; the lower series are scaled to Point 2, near the terminus. As in previous plots, the black lines in each series are process model output (ANICE-ITM for the example case) and the blue lines are stochastic realizations. Series share $x$ and $y$ axes.

to the SMB variability in space and time generated by the method described above. This stochastic SMB generation method
has been incorporated directly into the Ice Sheet and Sea-level System Model (Verjans et al., 2022).

In evaluating candidate stochastic generators for catchment annual mean SMB, we found the best fit to process-model variability with the lowest order statistical models. For all 260 catchments we tested, simple autoregressive models had by far the lowest Bayesian Information Criterion (better fit to process-model SMB) among model types (e.g. Figure 3b). Moreover, among low-order autoregressive models, white noise AR(0) models with a trend are preferred over higher-order models in
most basins, for all seven process models tested (Figure A1). Low-order AR models could have a low BIC despite relatively greater error than higher-order models, as seen in Figure 3, because the BIC penalizes excess parameters (Equation 2). For each process model, there are some basins where higher-order AR($p$) models are strongly preferred over white noise. The example application we have shown allows the best-fit model order to be selected per basin. Our workflow therefore provides a self-consistent way to infer stochastic generator fits for basins with different patterns of variability.

Our findings contrast with the results of a study by King and Watson (2020), which found that simple white noise and low-order AR models were not effective in capturing observed Antarctic SMB variability. For annual SMB time series, re-constructed for 1800-2010, for four Antarctic catchments–the West Antarctic Ice Sheet, East Antarctic Ice Sheet, Antarctic Peninsula Ice Sheet, and Antarctica as a whole– King and Watson (2020) used the software Hector (Bos et al., 2013a) to simultaneously fit a linear trend and noise model. They found that white noise and AR(1) models tend to underestimate low-

frequency variability, and that a better fit to observations came from power law or Generalized Gauss-Markov models (Bos et al., 2013b). The use of only 3 sub-catchments for Antarctica results in much broader spatial aggregation as contrasted with our use of 260 sub-catchments for the smaller Greenland Ice Sheet. That broad spatial aggregation might be expected to smooth short-term variability and amplify the relative importance of low-frequency variability that correlates with large-scale climate forcing. For that reason, it is not surprising that we find a better fit with simple temporal models given that we aggregate over

smaller ice-sheet catchments and study a shorter time period. Further, the spectra of variability could well be different between Antarctica and Greenland; the former is a polar continent with climate heavily influenced by the Antarctic Circumpolar Current, while the latter is a large subpolar island exposed to warm oceanic currents and westerly atmospheric flow. Antarctic SMB variability is thus dominated by snowfall (Previdi and Polvani, 2016), while Greenland experiences more surface melt and runoff, so the best-fit temporal model types may not be directly comparable. Finally, we have tested stochastic model fit

to more data sources — seven SMB process models — than did earlier studies of one or two data sources (including King and Watson, 2020); we found that simple autoregressive models were the best fit for all seven of the training models, lending credence to our results despite their contrast with earlier findings. We do expect the characteristics of the best fit stochastic generators to depend on basin delineation and training dataset, which we discuss further below.

We chose to limit the range of lags we tested in our autoregressive model fitting for two reasons. First, the autocorrelation

functions of annual SMB reconstructed from ice cores (Figure B1) show that most cores have significant autocorrelation at short lags ($< 10$ yr) and no consistently significant autocorrelation at longer lags. Second, higher-order autoregressive models risk both overfitting the data and needlessly adding computational expense, since high order autoregressive models require holding the SMB from many previous time steps in local memory. The Bayesian Information Criterion of candidate high-order AR($p$) model fits to SMB data is high in most basins, supporting our choice in this case. Further, the decadal timescale of

our example application is the most feasible timescale on which to generate probabilistic projections of sea level change. For timescales of 50 years and longer, uncertainty about anthropogenic emissions scenarios dominates the range of possible sea level change (Hinkel et al., 2019). However, it should be noted that a low-order autoregressive model such as ours is poorly suited to capture low-frequency variability, which may become important for multi-century simulations.

Ice core data (Figure B1) does not suggest that we have missed major modes of variability in our model fitting, but it is still

plausible that our stochastic generator fitted to 32 years of training data will fall short in reproducing multidecadal and longer variations. To ensure that stochastic SMB generators do not miss low-frequency variability that could substantially change Greenland outlet glacier catchments in the coming century, and to support stochastic generation for longer-term historical simulations, further analysis should incorporate longer-term process model output or spatially resolved reconstructions of SMB from ice cores or other observations. If the Greenland Ice Sheet were to become unstable, as recent analyses have suggested

(Boers and Rypdal, 2021), the variance and autocorrelation timescale of its future mass balance could be quite different from the recent past. Stochastic generator fitting intended for multi-century future projection should thus be trained on output data from SMB models run at similarly long time scales, where possible including the relevant feedbacks and instabilities, rather than projecting forward from 30-year historical simulations as we have done here. We emphasize that our study describes a flexible methodological framework for training a stochastic generator of SMB variability, with an example application to

multi-decadal simulation. Our framework can be applied to existing data for other use cases (such as paleo reconstruction) and to new SMB process model outputs as they become available.

Our downscaling method makes it possible to generate SMB fields on whatever mesh is needed by a numerical ice sheet model. In ice sheet models designed to accept stochastic forcing, the parameters of the stochastic generator can be provided directly for online generation of the forcing fields within the model itself, with negligible addition of computational expense

(demonstrated in Verjans et al., 2022). With regular updates to the surface elevation of each point on the model mesh, Equation 8 can also account for the known feedback between ice sheet surface elevation and surface melt rate (Hanna et al., 2013; Edwards et al., 2014; Lenaerts et al., 2019). Such a streamlined workflow will further facilitate large ensemble simulations.

The workflow we present here, including the downscaling method, is agnostic to choice of regions over which to aggregate the SMB. The example application to outlet glacier catchments in Greenland uses a standard, published basin delineation

(Mouginot and Rignot, 2019). The downscaled time series shown in Figure 3, which we generated with data aggregated that standard set of catchments, show variability dominated by large-amplitude seasonal variation at the terminus. This asymmetry in variability amplitude between the accumulation and ablation zones ultimately leads to some overestimation of interannual variability at accumulation zone points. When aggregated over a large accumulation area, overestimated local variability could translate to an artificially large magnitude of uncertainty in expected sea level contribution. We suggest that this effect could

be tempered by splitting catchment data into accumulation-area and ablation-area bins before fitting the spatial downscaling function. Depending on the user's scientific goal, such disaggregation may not be necessary for forcing an ice sheet model, as sub-decadal outlet glacier flow variability is driven by near-terminus SMB variability (Christian et al., 2020). We expect that there would be qualitative differences in the SMB series generated with and downscaled to different choices of basin delineation (Goelzer et al., 2020a); we have not attempted to optimize basin selection for the illustrative example here. Users

may apply all steps of the workflow described in Sections 3.1-3.6 to SMB data aggregated over different regions.

The within-catchment downscaling we present in Section 3.6 is a simple example that may be adapted or replaced for other applications. The example data plotted in Figures 6 and A3 show only 1980-1985 in the mass balance gradient. We tested example $\Lambda_\tau$ fits to data from the full period (1980-2012) but found that they tended to underestimate variability; conversely, fits to shorter periods tended to overestimate spatial variability. The example presented here illustrates the possibility of inferring

a downscaling function from process-model output. It would be possible to infer similar downscaling functions at different temporal or spatial resolutions, using reanalysis or reconstructed data, or computed over a different reference period. Ultimately, the choice of a reference period and the best spatial dataset to infer such a function depends on the user's intended application, and this selection may be non-trivial. Further, our simple downscaling does not capture changes in elevation dependence of SMB over time, for example due to changes in precipitation phase or local atmospheric lapse rate. Users seeking improved

fine-scale performance may wish to implement more granular statistical downscaling methods (e.g. Noël et al., 2016).

The inter-catchment spatial covariance method we apply here will lose some relevant spatial details from the original process models. As described in Section 3.3, the empirical inter-basin correlation matrices $\hat{\mathbf{C}}$ were singular for our example case, and in order to generate new realizations of variability, we enforced sparsity in the correlation matrix $\boldsymbol{\Gamma}$ (Figure 4a-b). By construction, this method loses some spatial detail present in the original dataset. Further, our method does not quantify uncertainty in the

405 model fit — for example, within-catchment differences in the best-fit statistical model parameters — other than the range of variability present in the original process-model simulations. Our stochastic generation of SMB fields based only on SMB models also disregards any covariance between oceanic and atmospheric forcings. More sophisticated methods currently under development, such as fitting a Gaussian process emulator (Mohammadi et al., 2019; Edwards et al., 2021) to the field varying in space, may be able to resolve these problems in the future. However, fitting such an emulator that varies in space and time

would require storage of, and computation on, multiple realizations of SMB process models at kilometer resolution. Such a task is considerably more computationally demanding than what we have pursued in the example shown here.

Given the simplifications described above, and the abstraction of stochastic parameters as contrasted with physical quantities, we do not intend stochastic SMB generation to completely replace process-model simulation of ice sheet SMB. Rather, we envision stochastic SMB generation to provide a complementary tool set which reproduces many features of SMB pro-

415 cess models at nearly negligible computational expense. The open source software that we have developed, and the existing packages on which it is built, can be easily applied to fit a stochastic representation to new outputs from process-based SMB models as they become available. Selecting an appropriate class of stochastic generator is the most time-consuming step of the process; with that complete, the best-fit model parameters can be updated at any time to account for new process model results, and generate hundreds of new realizations sampling the range of internal variability of SMB. Stochastic generation therefore

serves to more immediately connect dynamic ice sheet projections with internal variability from cutting-edge SMB simulations without the need for costly coupled ensemble simulations.

## 5   Conclusions

We have described the development and demonstrated the use of a stochastic method to generate many realizations of ice sheet SMB fields varying in space and time. For all 260 catchments of the Greenland Ice Sheet that we tested, the simplest temporal

models ($AR(p)$ with order $p < 5$) provided the best fit to process-model-derived SMB time series. Our method streamlines the creation of large samples of climate-dependent forcing to simulate ice sheet mass change subject to internal climate variability. The improved computational efficiency offered by this stochastic SMB generation method will facilitate large ensemble simulations of ice sheet change, which can support a range of applications including (1) probabilistic sea-level projections with improved uncertainty quantification, (2) separating ice sheet variability from atmospheric and oceanic variability in simulated

changes to the coupled climate system, and (3) attribution of observed changes to specific forcings.

*Code and data availability.*  Code supporting our analysis is available on GitHub (https://github.com/ehultee/stoch-SMB) and archived on Zenodo (doi: 10.5281/zenodo.8047501). Catchment-aggregated SMB time series derived from the participating models are included in a subfolder of our GitHub repository, github.com/ehultee/stoch-SMB/data.

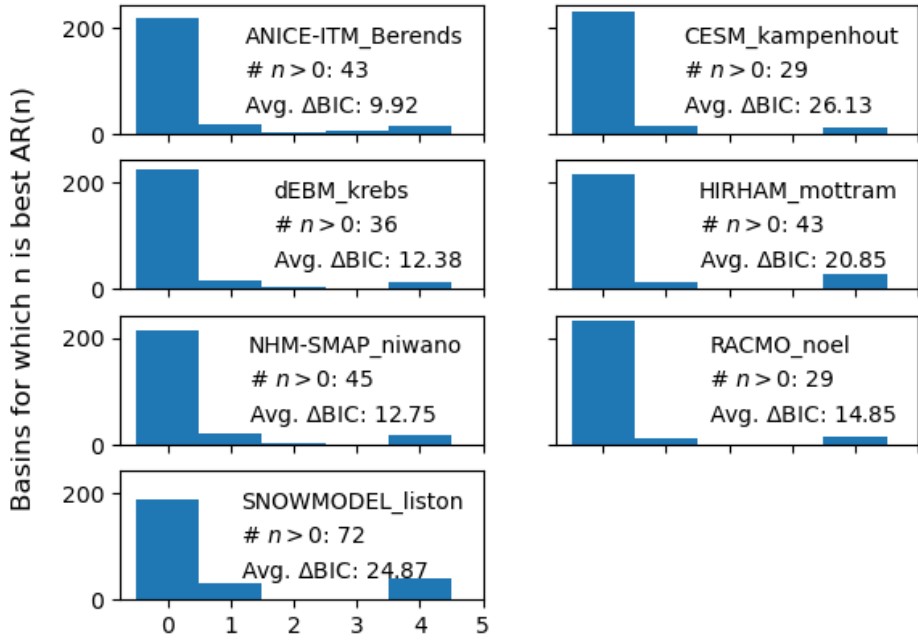

**Figure A1.** Histogram showing order $n$ of best-fit AR($p$) process, separated by process model. The number of basins for which the best fit $n$ is nonzero, as well as the average difference in BIC between the best fit and an AR(0) fit for those basins, is shown for each process model.

## Appendix A: Best-fit stochastic generators are similar for different SMB process models

The method we presented can be adapted for a variety of scientific applications. In the example use case we demonstrated above, we fit stochastic generators to the output from several SMB process models that had participated in the Greenland SMB Model Intercomparison Project (Fettweis et al., 2020, models described therein). The SMB process models vary in complexity, from relatively simple energy balance models such as ANICE-ITM (shown in the main text) to more sophisticated regional climate models such as RACMO.

The results of fitting a stochastic generator to the model output were comparable regardless of process model. Figure A1 shows how many basins were best fit by AR($n$) stochastic generators for $n$ from 0 to 5, with 0 being a white noise model. Figure A2 shows SMB series produced by stochastic generators fit to each of three example process models. The series are qualitatively similar; the amplitude of the variability in the stochastic realizations as compared to the process model series differs per process model. This effect is due to differences in the spatial covariance that was inferred from each process model

and used to generate the noise term $\epsilon(t)$ in each realization.

     Within-catchment spatial variation does differ slightly per process model. For example, the SMB spatial anomaly for points in the Kangerlussuaq Glacier catchment is different for ANICE-ITM (main text Figure 6) RACMO (Figure A3), especially at low elevations. RACMO shows less spread in January values and much more spread in July values, to the point of fitting an

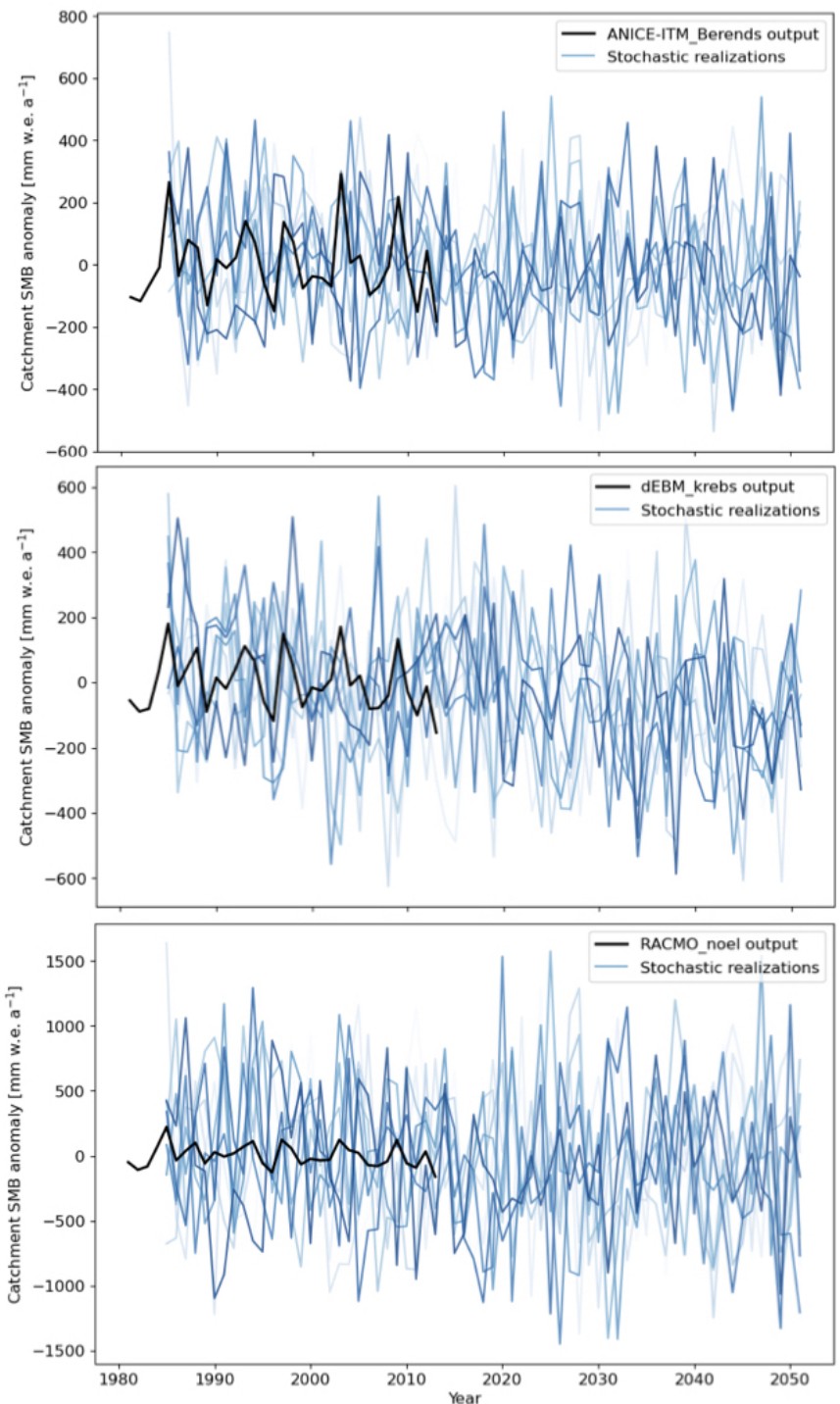

**Figure A2.** Example realizations from stochastic generators fit to three different process models: ANICE (top), dEBM (middle), and RACMO (bottom). The series shown is Kangerlussuaq Glacier SMB — as in main text Figure 5 — generated from 1980-2050.

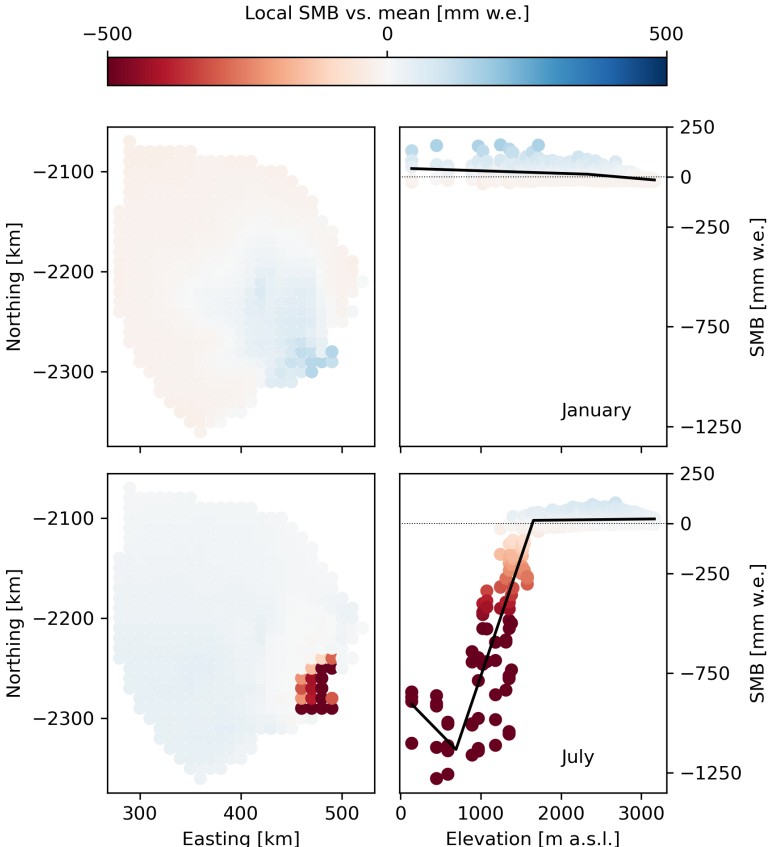

**Figure A3.** SMB downscaling in time and space, shown for January and July, as in main text Figure 6 but here showing fits to RACMO output rather than ANICE-ITM. Left panels: SMB spatial anomaly (difference from catchment mean) for each point within the Kangerlussuaq Glacier catchment, based on the RACMO model contribution to GrSMBMIP. Right panels: SMB lapse rate with elevation for each month, deduced from the anomaly fields shown. Colored data points represent individual pixel values, 1980-1985. Colormap is consistent across panels.

inflection point in the SMB-elevation relationship within the ablation area. Users of our method must determine what down-
scaling approach is most suitable for their scientific aims. Possible choices include (1) use a spatial downscaling consistent with the process model the user intends to sample, as we have shown in the main text for ANICE-ITM; (2) fit a monthly downscaling function comparable to Equation 7 but based on data from another source they find more accurate for this application, such as an observational dataset or a higher-resolution process model; or (3) implement another elevation-dependent downscaling technique such as those described in Noël et al. (2016) or Goelzer et al. (2020a).

## Appendix B: Modes of variability in ice core reconstructions

The GrSMBMIP process model output we used to fit stochastic generators in our example application covered a common period of 33 years, 1980-2012. To add longer-term context to our choice of candidate model classes (§3.2), we also examined ice core reconstructions of SMB in Greenland over the last 2000 years (Andersen et al., 2006). The point nature of these measurements makes them unsuitable for generating stochastic, ice-sheet-wide SMB fields, but they are a useful benchmark to assess the characteristic time scales of SMB variability, including time scales longer than are simulated in regional SMB models.

We computed the autocorrelation and partial autocorrelation functions of SMB reconstructed from each of five cores. If multiple cores showed significant autocorrelation at time lags longer than 5 years, it would be an indication that our model fitting procedure should include candidate models with higher autoregressive orders $p$ and moving averages $q$. The autocorrelation function for the ice core SMB is shown for lags up to 100 years in Figure B1. There are no lag values greater than 5 years for which the five cores agree on significant autocorrelation.

The ice core record in Andersen et al. (2006) comes from cores in the accumulation area of the Greenland Ice Sheet. The cores are not necessarily representative of decadal-scale SMB like the GrSMBMIP data we fit in our example application, because they will not reflect variation in melt rate or coastal precipitation. As a complementary data set, the ice cores support our choice to limit the lags $(p, q)$ tested in our model fitting procedure; however, they do not guarantee that our example SMB generators are applicable at time scales far beyond the historical period to which they were fit. For scientific applications that aim to generate SMB varying on time scales of centuries and longer, we encourage users to fit a generator to a training data set on a comparable time scale.

*Author contributions.* AR conceived of the Stochastic Ice Sheet project. LU and AR designed the SMB study, with support from SC. LU wrote the code, made the figures, and drafted the manuscript. All authors contributed to editing the manuscript and approved its final form.

*Competing interests.* The authors have declared that no competing interests are present.

*Acknowledgements.* The authors thank GrSMBMIP participating teams led by Tijn Berends, Leo van Kampenhout, Uta Krebs-Kanzow, Ruth Mottram, Masashi Niwano, Glen Liston, and Brice Noël for their permission to use their model data, and Xavier Fettweis for facilitating access to GrSMBMIP and MAR output data.

The authors thank Matt Osman for ice core data consultation, and Vincent Verjans for his work on $z$-score normalization for nonstationary ice sheet forcings. The authors also thank Tijn Berends and one anonymous reviewer for constructive comments that helped improve the manuscript.

This research has been supported by the Heising-Simons Foundation (grant no. 2020-1965).

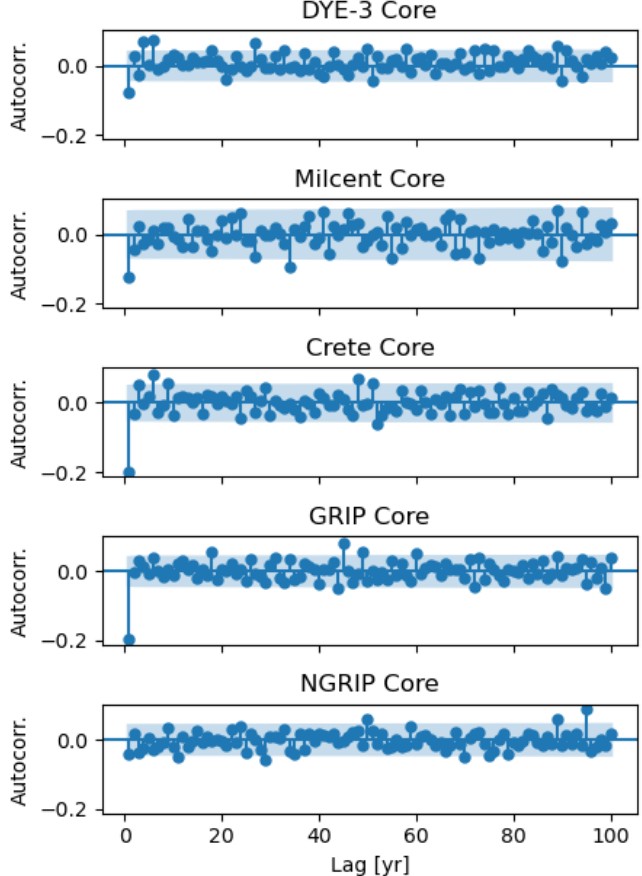

**Figure B1.** Example autocorrelation function of SMB derived from five Greenland ice cores, with time horizon out to 100 years. Shaded area shows the 95% confidence interval around 0, such that points outside the shaded area indicate autocorrelations that are significantly different from 0.

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
