# Peer review of "A stochastic parameterization of ice sheet surface mass balance for the Stochastic Ice-Sheet and Sea-Level System Model (StISSM v1.0)"

_EGUsphere, 2023_

## Author Response (AR1)

**Author Response to Reviews of**

**A stochastic parameterization of ice sheet surface mass balance for the Stochastic Ice-Sheet and Sea-Level System Model (StISSM v1.0)**

L. Ultee, A. Robel, S. Castruccio
*Geoscientific Model Development*
* * *
RC: *Reviewer Comment*,     AR: *Author Response*,     ☐ Manuscript text

We thank the reviewers for their thoughtful and prompt comments. We have made changes in response to each major point of the reviews. We respond to each suggestion directly below, where necessary quoting excerpts of revised text.

While revising, we corrected three coding mistakes that change some details of the example application, but do not change the overall workflow. For transparency, those changes are:

- Corrected a `for` loop that inadvertently left out several catchments of the Greenland Ice Sheet. We now present results for 260 catchments rather than the 200 discussed previously.

- Corrected our implementation of `statsmodels.tsa.ar_model.AutoReg` to use the optional argument `hold_back`. This argument, set to the maximum number of lags intended to be tested, ensures that AR models of different lags are computed on the same data and their BIC can be directly compared. This change truncates the length of the residuals used to compute the empirical correlation matrix. They are now 28 years long rather than the 30 previously stated.

- Corrected the test range of autoregressive lags to include 0 (white noise). We found that in many basins AR(0) is the preferred model; we comment further in the text and in our response to Reviewer 1 below.

**Summary of major changes**

- Revised throughout to address applications other than our 1980-2012 historical example — including the addition of a new Section 3.5: Non-stationarity

- Clarified the role of GrSMBMIP training data versus the complementary role of ice core data

- Given more description and illustration of model fitting procedure — including a revised Figure 2, a new Figure 4, a clearer Figure 6, and a new Appendix A

- Addressed the dependence on basin delineation throughout

**1. Reviewer 1**

**1.1. Summary**

**RC:** *The paper presents a method to synthetically generate surface mass balance for the Greenland ice sheet based on existing process model output. It allows to statistically produce SMB with the same overall temporal and spatial characteristics, but with different inter-annual variability. The parameterisation may be used to generate a large number of forcing to study the effect of internal variability.*

*I didn't review the provided example code in detail, but note that it is easy to access, well documented and facilitates reproduction of the presented material and the figures. This is a very nice example of open science. Thank you to the authors.*

**AR:** *We thank the reviewer for their attention. We appreciate their commendation of our open code. Open science is an important priority for us and motivated our choice of GMD for the manuscript submission.*

**1.2. General comments**

**RC:** *My main general comment is that the paper suggests (explicitly and implicitly, e.g. l3, l38, l58 and by the referenced papars) that the provided parameterisation of SMB could be used for future projections. However, the presented methods focus on stationary signals over the historical period. While we can imagine similar methods could be applied for projections, no evidence is given that this is actually the case. Additional considerations would certainly be needed to get there. This is mentioned in the discussion, but until then the reader may be mislead. I therefore suggest to either 1) make it clear from the beginning (abstract, introduction) that the given approach is only valid for SMB close to the historical, or 2) provide examples of an application for the multi-decadal to centennial times scale.*

**AR:** *Point well taken. We have clarified throughout the manuscript that we are showing an example application close to the observed historical period, and where necessary we have added discussion of what users would need to adjust in the workflow to construct generators applicable at other time scales. See e.g. line 10, lines 355-360, Section 3.5, and Appendices A-B. See also below response re: example application to century-scale projection in ISSM.*

**RC:** *I am concerned if the set of regions (Fig 1) is a meaningful and best choice for that type of analysis. The catchment delineation was motivated by the ice flow, not by SMB at all. The sizes of the catchments are all different, they encompass regions of largely different SMB, the relative share of accumulation vs ablation areas is different from region to region.*

*It is not immediately clear how an optimal delineation would look for SMB like instead, but it seems that something more adapted in at least some of the aspects above could be constructed. I am thinking e.g. of one of the higher order mascon delineations in https://doi.org/10.1093/gji/ggy242 or similar.*

*My main concern is in how much your results depend on that specific delineation. You mentioned that problem in in the discussion (l300), but did not provide further analysis about it. I think it would be useful to compare to at least one other definition of regions to show robustness of the results.*

**AR:** *We have added text addressing dependence on basin delineation. Ultimately, the choice of most meaningful basin delineation depends on the user's intended application; what we have presented is a workflow with a single example application from a wide set of possible applications. We prefer to keep the manuscript brief, focused primarily on documenting the workflow, rather than extending it to compare multiple example applications. Further, we examined the reference suggested by the reviewer but did not find a corresponding*

[Figure]

Figure R.1: Histogram showing order $n$ of best-fit AR(n) process, separated by process model. The number of basins for which the best fit n is nonzero, as well as the average difference in BIC between the best fit and an AR(0) fit for those basins, is shown for each process model.

*dataset of mascon outlines.*

**RC:** ***The model fitting process is not very transparent, because it relies on existing software packages and is not well described beyond that. I belive my confusing about the fitting process (comments l126 and l155) is related to that lack of description. I'd suggest to add more detail about how the fitting process works, what characteristics of the original time series are supposedly matched with the selected models and to illustrate these relationships with a figure if needed.***

 AR: *We have given more detail about the model fitting process, adding text where noted in the specific comments below. We have updated Figures 2 and 6 in the main text, added a new Figure 4, and added text and figures in a new Appendix A to give more context to the model fitting and generation.*

**RC:** ***And maybe a bit speculative, does any of the models significantly outperform a random (white noise) process scaled to the right amplitude per basin and model?***

 AR: *Yes — though in fact it was an oversight on our part not to include comparisons to white noise. We thank the reviewer for raising this question. White noise is preferred in many but not all cases, and where another model is preferred, the difference is non-trivial (average difference in BIC ranging from 10 to 26, for fits separated by model). We have added text and a supplementary figure (shown here as Figure R.1) to discuss this point.*

*We have also removed the focus on AR(1) as a universally acceptable fit. This point was undermined by the tests with white noise, and ultimately the point is not necessary for our example application nor for users of the workflow.*

**RC:** ***Also, while I can see how the noise characteristics may slightly differ and match better or worse, how do projected mass changes produced by an ice sheet model under these different forcing attempts compare.***

*Since this method is already integrated in ISSM, could you show some results? How does your parameterisation with the best statistical model compare to sampling individual years randomly, which is what people have tried in the absence of an appropriate statistical tool?*

AR: *This is a very insightful suggestion, but it goes beyond the scope of the current manuscript. We have added a citation to Verjans et al. (2022) in the introduction for more in-depth description of the implementation within ISSM. In this manuscript, we present a method of constructing a stochastic generator that can be adapted for several possible use cases. The example we have demonstrated, fitting stochastic generators to SMB process model output from a historical period, could be applied to attribution of sources of uncertainty during the observed historical period. Future projection (or paleo reconstruction) is another use case and, as both reviewers note, would require a different training set with additional processing to construct a suitable generator. We have added text in lines 240-252 and 458-462 to clarify this. In a forthcoming study, we apply the method to construct a generator on longer-term training data for century-scale projections.*

**1.3. Specific comments**

RC: *Title: Given that the presented parameterisation is specifically developed for the Greenland ice sheet, I would suggest to add 'Greenland' in the title. Clarifying this may also be important in light of contrasting results found from Antarctica as discussed.*

AR: *The method is applicable to either ice sheet; the example application is Greenland. We appreciate the reviewer's suggestion but prefer to keep the description general. We have revised the abstract and the main text to clarify that we present an ice-sheet-agnostic method with an example application to recent Greenland SMB.*

RC: *l1 "Many scientific and societal questions that draw on ice sheet modelling could be best addressed by sampling a wide range of potential climatic changes and realizations of internal climate variability" — I can think of many questions that were better addressed by focussing on a specific scenario or where internal climate variability is not that relevant. Suggest to remove the superlative "best" from this sentence.*

AR: *Changed wording to "...necessitate sampling a wide range...". We had not intended the interpretation as a superlative. Thanks.*

RC: *l5 "The wide sampling required to address such questions is computationally infeasible with sophisticated numerical climate models at the resolution required to accurately force ice sheet models. Stochastic generation of climate forcing of ice sheets offers a complementary alternative." — The presented parameterisation is limited by the available high-resolution climate model simulations for a specific time-period and scenario. The added value is mainly in generating internal variability, which is an important but not sufficient ingredient to produce a wide range of forcing. The limitations of the method should be recognised and made clear in the text.*

AR: *Revised throughout to specify that stochastic generators help sample internal variability, and that different training data will be necessary for different scientific use cases.*

RC: *l11 "while including feedbacks to ice sheet surface elevation" — More than one feedback? It is basically the SMB-height feedback. Reformulate.*

AR: *Changed to "while including feedback from changing ice sheet surface elevation."*

RC: *l38 "providing too little information to estimate the probability distribution of output variables such as future sea level" — This is the reason process-based ice sheet sea-level projections had to be emulated for*

*the AR6 assessment. Could be added with a reference to Edwards et al. (2021).*

AR: *Added a short paragraph highlighting this point and citing Edwards et al. (2021). Thanks!*

RC: *l43 "ice sheet models are not typically forced directly by climate model output." – Suggest to add "global" before "climate model output" to distinguish between GCMs/ESMs and RCMs. The latter also being climate models that are used to force ice sheet models directly.*

AR: *Added, good point.*

RC: *l44 Maybe mention "downscaling" here and be more specific. "Rather, global climate model output must be downscaled to construct an SMB field of high enough spatial resolution and quality, often through use of a specialized mass and energy balance model that accounts for processes at the snow/ice surface and in the snowpack".*

AR: *Revised to exactly the suggested wording. Thank you!*

RC: *l48 "Stochastic methods provide a low-cost alternative to sophisticated process models". — I find this sentence somewhat in contrast to l292 " ... this framework can be applied to new SMB process model outputs as they become available". While l48 can be read to mean that stochastic methods can replace sophisticated process models altogether, l292 recognises that your method is based on available process model results. Suggest to reformulate l48.*

AR: *Revised to clarify that stochastic generators offer "a low-cost alternative to **ensembles with multiple realizations of** sophisticated process models."*

RC: *l54 Suggest to distinguish between the generator and the SMB product it generates. The latter is what the 'which' in the sentence can refer to: "to construct a statistical generator that can produce a SMB to force an ice sheet model, which should include ..."*

AR: *Revised to "...to construct a statistical generator of SMB to force an ice sheet model. The SMB product we wish to generate should include..." Good catch.*

RC: *l56 Wouldn't the sentence make more sense when 'a' and 'one' are exchanged? "We approximate the output of one process-based SMB model as a realization of a stochastic process"*

AR: *Revised as suggested.*

RC: *l56 "The statistical model that produces a realization best fit to the process-based model output can then be used to generate hundreds of other realizations" — To get the concept clear already here: Do you produce one statistical model for each process model, or one statistical model that generalises over an ensemble of process models? Clarify. Maybe "produces a realization best fit to a process-based model"*

AR: *Revised to read:*

> *The statistical generator that produces a realization best fit to a given process-based model output can then be used to generate hundreds of other realizations, representing a range of possibilities for future SMB consistent with the same model, at much reduced computational expense.*

*It is one statistical model ("generator") for each process model, as should be more clear now.*

RC: *l58 "representing a range of possibilities for future SMB" — Most of your examples focus on the historical*

*period and it is not clear how that extends e.g. to a future under strong warming. From looking at available RCM SMB until 2100, it seems that the amplitude of inter-annual variability increases already by 2050. How will this be incorporated into the statistical model?*

AR:    *This is a great point, and we have indeed encountered this increased amplitude of inter-annual variability as we construct generators fit to forward-projected forcing data (current work). We have added text to address use cases with non-stationary variance in a new Section 3.5:*

> *The GrSMBMIP process model historical output we use as our example application did not exhibit non-stationarity, according to the KPSS and Augmented Dickey-Fuller tests applied to each output series (§**??**). We therefore fit stochastic generators that were stationary by construction, assuming the underlying distribution of the data did not change over the period of simulation. We generated stochastic forward simulations as shown in Figure **??** to illustrate the possibility of generating time series with consistent variability outside of the training period. Those simulations fit a linear trend to the training data and assumed that the trend and amplitude of variability remained constant into the future. For scientific applications that study periods of varying climate — for example, glacial-interglacial periods, or century-scale climate projections with anthropogenic forcing — it is expected that SMB time series would not be well fit by stationary models (Weirauch et al., 2008; Bintanja et al., 2020).*
>
> *To fit a stochastic generator to time series with statistics (mean, trend, variance) varying over time, the user could subdivide the training data series to periods with stationary trend and variance. Piecewise linear trends could be computed on the sub-series, and each series normalized by its variance to create a "z-score" time series. The stochastic temporal model could then be fit to the z-score time series as described above. The output of the stochastic generator would then be re-scaled by the variance in each period to produce ensembles of non-stationary SMB series. The best choice of break points to sub-set the training data will depend on the user's priorities and the time period being investigated; we do not pursue z-score rescaling any further in this example. For more formal discussion of bias correction in the case of climate data whose distribution changes in time, we refer the interested reader to applied statistics literature e.g. Zhang et al. (2021) and Poppick et al. (2016).*

RC:    *l60 "we base our method entirely on open-source software packages and provide our own open-source code where necessary." — That is very well appreciated. Thanks again.*

AR:    *Thank you!*

RC:    *l62 add "surface": "stochastic surface mass balance generator"*

AR:    *Added.*

RC:    *l68 "First, we will examine ice core reconstructions of SMB in Greenland" — I am missing more detail and I am also concerned about the usefulness of ice core data for this study in the first place. Where are these ice cores located? What kind of information do they provide (before processing autocorrelations)? If they are from regions in the accumulation area, the constraining potential is probably limited to large scale snowfall. Can they be assumed representative for the SMB you are interested in, which may instead be dominated by variations in melt and coastal precipitation? I believe a critical evaluation and better documentation of this data source is important.*

AR:    *This point is very well taken. We have revised to describe the ice cores as a complementary dataset, and we have added an appendix with more discussion of the dataset and its limitations. We had intended to discuss*

*the ice cores in more detail but we see that did not make it into the previously submitted manuscript; we thank the reviewer for pointing that out.*

**RC:** ***l75 Not sure what makes them "benchmark SMB models". Maybe remove "benchmark". Consider to also list the model names here.***

*AR:* *Revised as suggested. Text now reads:*

> *The subset of GrSMBMIP models we analyse comprises those whose developer team gave us permission to use their archived data for this purpose: ANICE (Berends et al., 2018), CESM (van Kampenhout et al., 2020), dEBM (Krebs-Kanzow et al., 2021), HIRHAM (Langen et al., 2017), NHM-SMAP (Niwano et al., 2018), RACMO (Noël et al., 2018), and SNOWMODEL (Liston and Elder, 2006). This selection includes exemplars of simpler energy-balance models as well as more sophisticated regional climate models (Fettweis et al., 2020) and these models have been extensively validated against observations over recent decades.*

**RC:** ***l80 "catchment outlines (Figure 1) provided by Mouginot and Rignot (2019)" — I am concerned about the choice of region delineation. See general comment.***

*AR:* *Revised to address basin delineation in the discussion. See response to general comment above.*

**RC:** ***l80 "Delaunay triangulation to produce a covering of each catchment area" — I don't understand the need and purpose to triangulate the domain. Would it not be enough to select and sum/average over the 1km grid cells that fall into a specific catchment area?***

*AR:* *Revised to "sum the grid cells that fall within each catchment area, dividing by the total area of the catchment to arrive at catchment mean SMB for each month, catchment and model from 1980 to 2012." The reviewer is correct that at 1-km resolution, a naive summing of grid cells whose centers fall within basin boundaries should be enough. We have confirmed this in our subsequent work on large-ensemble simulations with ISSM forced by stochastic fields (different manuscript). In the example shown here, we applied Delaunay triangulation to account for grid cells that fell only partially within basins — but the specifics of the summing method are not relevant to the workflow, so we have removed the mention here.*

**RC:** ***l81 "We average SMB over the grid point closest to (or within) each triangle of a catchment". — Does each catchment have one triangle or several? Sorry I don't understand the process. Can you explain this better?***

*AR:* *See above. We have removed the mention of Delaunay triangulation, which we agree was poorly explained and not relevant to the main workflow. The user only needs to create a catchment mean SMB, which can be accomplished through naive summing.*

**RC:** ***l82 "We then average over annual time scales" — Is this a moving average or just summing monthly values to get annual SMB?***

*AR:* *It is annual SMB, spatially averaged over the catchment. Revised:*

> *We average SMB over the grid point closest to (or within) each triangle of a catchment to arrive at catchment mean SMB for each month, catchment and model from 1980 to 2012. We then **sum to annual time scales, and divide by the total area of the catchment,** so that the subsequent analysis produces statistical models of inter-annual variability.*

**RC:** *l92 Maybe you base this on formal mathematical theory, but I found calling $\varepsilon(t)$ here an "error term" confusing. Since you describe the general form of the equation, maybe something similar to l106 "spatial correlations between catchments are captured in $\varepsilon(t)$" would be better here.*

*AR:* *We have switched to calling it a noise term throughout, which is also more consistent with our mention of "spatially informed noise generation" in section 3.3. Thank you.*

**RC:** *l96 How does m=30 map to the 33 year data set 1980 to 2012?*

*AR:* *Corrected to $m = 28$, which is the length of the 33-year time series with 5 years held back. This is required to make comparable AR(n) fits of lags 1 through 5.*

**RC:** *l97 This sentence is confusing to me. Is f(t) or b1(t) the forcing variable? — Maybe, if that is correct: "The temporal trend (t) includes historical mean SMB for each catchment 0, and the forcing variable f(t) with a linear coefficient 1. The forcing variable, f(t) ..."*

*AR:* *Yes, correct, revised as suggested.*

**RC:** *l105 "The error term (t)" — See comment l92, maybe call it something else.*

*AR:* *See above - switched to "noise term" throughout.*

**RC:** *l117 ARFIMA is both used to refer to the family of models (AR, ARIMA, ARFIMA) and to a specific choice. This should be resolved to avoid confusion.*

*AR:* *Revised this paragraph to disambiguate:*

> *These criteria guide our investigation of three common types of **temporal models. All temporal models we test belong to the** autoregressive-fractionally integrated moving average (ARFIMA) **family of** models. ... To avoid confusion, we henceforth use "ARFIMA" to refer only to ARFIMA models that do include non-integer differencing d, and we refer to the special cases ARIMA and AR(p) by their own names.*

**RC:** *l125 I don't see what choice for f(t) is used here. Is this one of the parameters that are optimized?*

*AR:* *Clarified in line 141: " In each case, we assume a linear dependence on time, $\beta_1 f(t) = \beta_1 t$ in Equation 1b."*

**RC:** *l126 I am confused about the model fitting process and request some clarification. The SMB time series have a certain inter-annual variability pattern in time. Is the purpose of model fitting to fit that specific time series or to find a model that can produce a time-series with the same characteristics (maybe amplitude, frequency)? I would assume the latter. If so, how do you characterise the produced time series? Did you look at the range, amplitude distribution and spectrum? The example fits shown in Figure 3 have similar time evolution, but that may not be the criterium used for a model selection.*

*AR:* *We have expanded our description of the fitting procedure in section 3.2. We added the following text:*

> *We analyse the Bayesian Information Criterion (BIC) as returned by the statsmodels built-in function for the temporal models fit to each catchment series. The BIC is given by:*
>
> $$BIC = -2\ell + \ln(T)(1 + df), \qquad (1)$$
>
> *where $\ell$ is the log-likelihood function of the given temporal model on the data, $T$ is the number of observations, and $df$ is the number of degrees of freedom in the generator. Minimizing the BIC balances a maximization of log-likelihood $\ell$ — the probability that a stochastic generator of this type could have produced the data series from the process model being fit — with a penalty for excess parameters (overfitting). We select the temporal model with lowest BIC for each catchment, for each SMB process model. We analyze the preferred temporal model types across all catchment-model pairs to identify the most suitable class of temporal models. We chose to select for minimum BIC to encourage computationally cheap models with fewer parameters (as in King & Watson 2020); we note that statsmodels also returns other common metrics of model fit such as the Akaike Information Criterion, which could be selected by users with other priorities.*

*To respond directly to the reviewer's question: a generator that produces series with amplitude and frequency most similar to the training data should also maximize log-likelihood $\ell$.*

**RC:** *l155 This seems like a surprising result and conclusion to me. How can the AR(1) process be a good choice, if the amplitude of the variability is (guessing from the figure) a factor 20 lower than expected? Wasn't the aim of finding a good model to be able to reproduce the inter-annual variability of the process model? If "residual variability may be captured in the spatial noise", why bother finding a model for inter-annual variability at all? Then the AR(1) process could maybe be replaced by a constant and all the variability captured in the spatial noise?*

*AR:* *You're right! We revised to remove this conclusion as it was poorly supported and unnecessary; see above discussion of AR(0) white noise as well.*

**RC:** *157 Isn't it a problem for the covariance analysis that the catchments all have different sizes and shapes and the statistical SMB is fully correlated on the catchment level? This is related to my general comment about optimal delineation.*

*AR:* *Revised to address this point in lines 194-196. The statistical model assumes a catchment-specific variance, in order words the SMB is assumed to vary differently for each catchment to implicitly account for different sizes. The spatial model is instead defined for the correlation, which we assume not to depend on the catchment size.*

**RC:** *l165 How does the method get information about where the basins are spatially located and if they are close to each other or not? Are neighbouring basins now more likely to see the same SMB in one year? Can basins that touch along the divide be considered neighbours in this approach.*

*AR:* *We have clarified in the text that the structure of the covariance matrix is inferred from the SMB process model data, not imposed from physical principles. Basins that covary in the SMB process model data will be more likely to covary in the generated realizations — whether because they are close to each other or for another reason.*

**RC:** *l166 "We calculate the empirical correlation matrix C^ , which is an approximation of C, from the residuals of per-catchment best-fit models described in Section 3.2" — The sentence seems to suggest that each catchment has one time vector of residuals. This goes back to my question above (l126). Ultimately each*

*statistical model can produce a number of realisations with different inter-annual variability. Wouldn't each of these produce another residual. Is the best-fit model one realisation or the model that can produce the best realisation among the infinite number of possible cases?*

AR:  *The reviewer's question highlights an important point of clarification. There are two parts to what we present: in the first part, we infer the form of the statistical generator and spatial noise from the training data provided by SMB process models; in the second part, we generate realizations from the generator we have constructed.*

*First, the best-fit temporal model is found by inferring its parameters in the fitting process described above. The residual — the difference between the SMB process model data and the best-fit model as fit through this process — is unique by construction, and that is what we use to compute the empirical correlation matrix $\hat{C}$.*

*The possibility of infinitely many realizations of $M(t)$ (Equation 1a) comes in the generation phase, using \*both\* the AR(n) process $\mu(t)$ and the spatially informed noise $\epsilon(t)$.*

RC:  **l169 Not sure showing the code here is useful, given that everything else until now is described in mathematical notation. I would prefer to continue in that way and maybe move the code to an appendix if it is deemed important.**

AR:  *Agreed. We have revised the text to better describe the matrices and removed the code snippet, which we determined was redundant.*

RC:  **l187 What is missing in the description for me here is a visualisation of the effect of the spatial correlation in the model. Could you add a figure.**

AR:  *We have added a new Figure 4 (shown below as Figure R.2), which shows the empirical correlation, the sparse correlation, and a map view of the correlation with a single example catchment.*

RC:  **l192 I agree with this interpretation, except for the part "for ensemble simulations of ice sheet change given uncertain SMB". Uncertainty in SMB (in particular in the future) is due to many factors aside from internal variability. The method is useful to produce/introduce inter-annual variability, but needs to rely on existing SMB for a given scenario, GCM and downscaling.**

AR:  *Replaced "uncertain SMB" with "SMB subject to internal variability."*

RC:  **l202 A similar catchment-based elevation-SMB relationship has also been used in Goelzer et al., 2020b: https://doi.org/10.5194/tc-14-1747-2020. Could add a reference here.**

AR:  *Thanks! Added with the other references in "previously applied in deterministic frameworks".*

RC:  **l205 "generally quite good for Greenland SMB" — It could be good to mention that the SMB-elevation relationship is very strong in the ablation area and lower accumulation area, but mostly breaks down in the upper accumulation area. This is because melt is fundamentally controlled by temperature, while snow accumulation is also controlled by wind. This is the reason why the SMB-height relationship in ISMIP6 was based on runoff gradients instead of gradients in SMB itself.**

AR:  *Interesting point, and the physical argument makes sense, but it is not what we see in the example data we present. The spread in SMB among points at comparable elevation, or among different years sampling the same point, is as large or larger in the ablation area as in the upper accumulation area of Kangerlussuaq. We also consulted the ISMIP6 experimental protocol in Nowicki et al. [1] and did not find a matching description of the SMB-height parametrization. No action taken here.*

RC:  **l208 "to capture the known feedback between ice sheet surface elevation change and SMB change" —**

[Figure]

Figure R.2: Illustration of inter-catchment covariance in ANICE-ITM output data. Panel a: the empirical correlation matrix $\hat{C}$, computed as described in §**??**. Panel b: the sparse covariance matrix that results from applying `GraphicalLassoCV` to $\hat{C}$. Panel c: the first row of the sparse covariance matrix (line 0 in panel b) translated to map view. Catchment 0 is Umiammakku Isbræ, indicated on the map with "UMI" and an arrow to its terminus. All panels share the colorbar shown below panel c.

*Equation 7 (and 5) do not capture that feedback. I think in its present form it produces information only at the same elevation as the process model. It should be explained how the equation needs to be adapted to map the SMB to the evolving ice sheet geometry*

AR: *Good point. We have revised our notation to define the monthly mass balance gradients as functions of elevation rather than position, and we have clarified that the elevation used to construct the local mass balance a may evolve in time.*

**RC:** ***l214 The potential problem I see with this approach in a non-stationary situation is that the number of months/years used to construct the gradients could be rather small. At least there is a conflict between producing robust gradients and capturing the transient SMB change with time. This issue may be raised here.***

AR: *We have added a paragraph discussing the downscaling in the discussion. We address this point and your later one about temporal resolution there.*

**RC:** ***l211 'SMB anomaly' is an often used term to describe the SMB difference relative to a reference climate. Here it is a difference of the local SMB to the catchment mean SMB. Suggest to make that explicit as: "the local SMB anomaly relative to the catchment mean". Same in l223***

AR: *We have revised to call it the "local SMB spatial anomaly". We give an explicit definition in Equation 6, immediately following the first use of the term.*

**RC:** ***l224 'The same principle could be adapted for training data provided at even finer temporal resolution (i.e., weekly or daily).' — I think this statement is too easily made, it needs some additional work. It is not obvious in that case how to build the gradient lookups. If we go for daily time resolution, selecting all January 1st and June 23rd would probably produce very inconsistent gradient maps, because of the weather in the system. Finding the right averaging period and enough data is not easy in that case. See also comment l214.***

AR: *Good point. We have added a paragraph to the discussion to address data selection and limitations of the method.*

**RC:** ***l256 Why did you not test white noise as an option?***

AR: *Oversight on our part! We have gone back and added white-noise comparison throughout. It is often preferred, reinforcing our emphasis on simple models – but in about 15% of the basins there is a strong preference for higher-order AR models, which supports the need for a self-consistent means of inferring stochastic generator fits for basins with different variability.*

**RC:** ***L266 In addition, Antarctic SMB variability is dominated by snowfall (hardly any runoff) and therefore quite different from Greenland. If you were to fit a large central accumulation area region in Greenland instead of the elongated drainage basin regions, maybe results would be more comparable to findings for Antarctica? That is something you could easily test to maybe add some substance to this more speculative paragraph.***

AR: *Thank you - we have added this point in lines 339-344. We have not tested other example basin delineations but have discussed them at length in lines 374-386.*

**RC:** ***269 "seven SMB process models" — I am missing discussion of the differences between those seven models. Did emulating them with the statistical model produce similar results?***

AR: *Yes, we found similar results for all seven models. We have added a short appendix with explanatory text*

*and figures to the manuscript. See figure R.1 in this response document and figures A1 and A2 in the revised manuscript.*

**RC:** ***277 "the decadal timescale on which we focus" — This does not become clear from reading the manuscript until here and is also in contrast to l330. If it is true, I think this sentence should be in abstract and introduction to clarify the focus of the paper.***

*AR:* *We have revised throughout, including abstract and introduction, to specify what is essential to the method for constructing generators, and what is specific to our example application on decadal-scale SMB from Greenland.*

**RC:** ***287 "Recent analyses also suggest that parts of the Greenland Ice Sheet may be approaching a destabilization" — I am not aware of any evidence for tipping behaviour on that time scale aside from this one paper. So I would be hesitant to propagate it.***

*AR:* *Good point. We have rephrased to indicate that this is an important consideration \*if\* such destabilization emerges:*

> *If the Greenland Ice Sheet were to become unstable, as recent analyses have proposed (Boers and Rypdal, 2021), the variance and autocorrelation timescale of its future mass balance could be quite different from the recent past*

**RC:** ***l315 "there is too little data available from high-fidelity models of SMB to pursue spatial methods beyond what we have shown here" — I think there is quite a range of process model output available by now and suggest to make a statement more humble to potential other research in this area.***

*AR:* *Revised to clarify that the limitations are not only data availability, but also storage and computational power.*

> *More sophisticated methods currently under development, such as fitting a Gaussian process emulator (Mohammadi et al., 2019; Edwards et al., 2021)* **to the field varying in space***, may be able to resolve these problems in the future. However,* **fitting such an emulator that varies in space and time would require storage of, and computation on, multiple realizations of SMB process models at kilometer resolution. Such a task is considerably more computationally demanding than what we have pursued in the example shown here.**

**RC:** ***l325 What do you mean by "coupled simulations"? Maybe "Regional Climate Model simulations"?***

*AR:* *Revised to:*

> *the best-fit model parameters can be updated at any time to account for new process model results, and generate hundreds of new realizations* **sampling the range of internal variability of SMB***. Stochastic generation therefore serves to more immediately connect dynamic ice sheet projections with* **internal variability from** *cutting-edge SMB simulations without the need for costly coupled* **ensemble** *simulations.*

**2. Reviewer 2: Tijn Berends**

**2.1. Summary**

**RC:** *Future projections of the sea-level contributions of the Greenland and Antarctic ice sheets depend on accurate projections of the surface mass balance of these ice sheets. The atmospheric processes responsible for the surface mass balance are affected not just by long-term climate change, but also by natural variability. As both observations and model simulations of the atmosphere contain signals from both long-term trends and stochastic variability, separating the two is problematic.*

*The authors present a novel method for efficiently generating many realisations of the surface mass balance, using a stochastic parameterisation. The parameters of this model are fit to realisations of different process-based SMB models, so that the statistical properties of the results from the process models are reflected in the results of the stochastic model. This allows ice-sheet models to run large ensembles of simulations, exploring the phase-space of SMB variability without having to run the computationally expensive process-based SMB models.*

*I think this an interesting and valuable development. In general, I think the authors present their methods and results well. Being an ice-sheet modeller myself, my knowledge of statistical mathematics is insufficient to have an informed opinion about the details of the stochastic model presented here. However, I have some more general questions that I think could be answered in the manuscript.*

**AR:** *We thank the reviewer for their assessment. It's very useful to have this perspective from an ice-sheet modeller who developed one of the SMB process models we analysed!*

**2.2. General comments**

**RC:** *1) Relative contribution of uncertainty in SMB.*

*In ISMIP6 and related projects focusing on future projections of ice-sheet mass loss, there is a general tendency to focus either on exploring the uncertainty in the ice-sheet models themselves (poorly constrained physical quantities and processes of the ice-sheet, model numerics, resolution, etc.), or on exploring the uncertainty from the climate/SMB forcing, which is usually done by using different CMIP models as forcing. It would be interesting to see how large the contribution from variability in SMB projections to the uncertainty in sea-level projections is, compared to the uncertainties already explored in ISMIP6. I think there is already some work on this in the Verjans et al. (2022) StISSM paper, and I think it would be informative to devote a bit more text to this in the introduction.*

**AR:** *Good point. We have added several sentences addressing this in the introduction.*

**RC:** *2) Relevance for ice-sheet modellers*

*The stochastic models presented here are trained on different members of the GrSMBMIP model ensemble. The authors mention that an advantage of their stochastic model is its low computational cost, allowing for large ensembles of simulations. However, about half of the models in the GrSMBMIP ensemble are also relatively cheap in terms of computation time (specifically the energy balance models and the positive degree day models), to the point where they would not noticeably slow down an ice-sheet model. What exactly is the advantage of using the stochastic model over any of those "cheap" SMB models? Does it allow for a more comprehensive sampling of the SMB phase space? Does it produce a better agreement with the RCMs like RACMO or MAR (which are generally agreed to provide much more accurate output than the "cheap" SMB models)? I think the relevance of the stochastic model could be explained better.*

*AR:* We have revised to clarify that stochastic generators are an alternative to **multiple realizations of** SMB process models; that is, they facilitate more comprehensive sampling of SMB subject to internal variability. See lines 56-58, 68-69, 410-414, and response to Reviewer 1, above.

We have also added a short appendix to clarify that the stochastic generator results are similar for the EBMs and RCMs we tested. Thus, stochastic sampling can provide a cheap alternative to the expensive RCMs as well as the "cheaper" ANICE that we showed in the previous version.

**RC:** *3) Downscaling*

**The authors present a relatively simple downscaling scheme to convert the catchment basin-averaged SMB time-series produce by the stochastic model, to 2-D data fields that can be used in an ice-sheet model. The problem of down-scaling low-resolution (climate) model output to a higher-resolution ice-sheet model grid is not new, and also not entirely trivial. Right now, the only evaluation of the downscaling method presented here, is a visual inspection of output for a single catchment basin. The timeseries in the top panel of Fig. 6B suggest that the stochastic model greatly overestimates the variability in the accumulation at this location, at least when compared to the process-model output (is the process model here ANICE-ITM again? If so, mention this – and why you chose to compare to this one, instead of the much more accurate RACMO or MAR data). I'd like to see a bit more analysis of the performance of this downscaling method. How does the spatial variability in the resulting 2-D field compare to e.g. RACMO? How does this compare to existing downscaling methods, e.g. Noël et al. 2018?**

*AR:* We have added a figure and text in Appendix A to describe differences between ANICE-ITM (shown in the main text) and RACMO. We have added references to Noel et al 2016 and Goelzer et al 2020 to recommend other downscaling approaches. Ultimately, we believe the novelty of our method is primarily in the construction of the stochastic generator for catchment-mean SMB; any downscaling method that can convert a catchment mean to a local SMB could be applied to the output of the temporal generator. We have clarified this in lines 380-393.

**RC:** *4) Long-term trends*

**It is not clear to me from the manuscript, how the stochastic model deals with long-term trends in projected SMB. In Fig. 4, ten different realisations of the stochastic model are compared to the ANICE-ITM output (again, why not compare to RACMO or MAR?). That runs to 2014, whereas the stochastic models continue to 2050. Does this mean the trend in the "observed" SMB (which ANICE-ITM, and the other models in GrSMBMIP, attempt to reproduce) is extrapolated? How would that work when I want to simulate ice-sheet mass loss under different RCP scenarios? As this is ultimately the main goal of ice-sheet models, I think this deserves some attention here.**

*AR:* We have added text in Section 3.5, quoted above in response to Reviewer 1, to describe a procedure for generating SMB series with non-stationary mean and/or variance.

The GrSMBMIP dataset was a useful test case because it included many different kinds of models, all with the same spatial and temporal resolution. With the demonstration of projections to 2050 in Figure 6, we simply intended to show that the stochastic model could produce realistic values outside the domain of the training dataset. There are indeed many use cases where a trend assumed constant in time would not be suitable. Thank you for pointing out that this was unclear.

**2.3. Specific comments**

**RC:** *Line 80: "We overlay ... and use Delaunay triangulation to produce a covering of each catchment area." I'm unclear what you mean here. As far as I know, the Delaunay triangulation is a unique triangulation for a given set of points in the plane. How does this relate to the catchment basins shown here?*

*AR:* *We have removed this mention of Delaunay triangulation. It was an irrelevant detail, and we agree it was confusing to present it as we did. See response to Reviewer 1 as well – the user only needs to create a catchment mean SMB, which can be accomplished through simply summing the cells that fall within the catchment basin boundaries. Sentence now reads:*

> *We overlay each field with the catchment outlines (Figure 1) provided by Mouginot and Rignot (2019) and* **sum the grid cells that fall within each catchment area, dividing by the total area of the catchment** *to arrive at catchment mean SMB for each month, catchment and model from 1980 to 2012.*

**RC:** *Figure 5: it is not immediately obvious what the small inset panels mean, until you notice that only the first one has tick values (but no axis label). Maybe consider only showing one or two months (I think that would be enough to get the point across), and promote the inset panels with the downscaling lapse-rate fit to full panels?*

*AR:* *We have replaced Figure 5 with a much cleaner version according to the reviewer's suggestion, showing full panels of the map and the lapse rate fit for only January and July. Great suggestion, far more readable. Thank you!*

**RC:** *Lines 303 – 306: "We suggest ... near-terminus SMB variability." This is an oversimplification. Overestimating the accumulation in the interior will not immediately affect the flow of the outlet glaciers, but it will strongly reduce the projected sea-level contribution of the ice sheet (this is especially apparent in the ISMIP6 Antarctic projections, where many ensemble members show a net mass gain due to increased snowfall in the East Antarctic interior).*

*AR:* *Revised to make clear that it is the \*variability\* in SMB that is being overestimated, and that it could lead to artificially large uncertainty in the sea level contribution. Thank you.*

**References**

[1] S. Nowicki et al. (2020), *Experimental protocol for sea level projections from ISMIP6 stand-alone ice sheet models. The Cryosphere, 14(7): 2331–2368. https://doi.org/10.5194/tc-14-2331-2020*

---

## Author Response (AR2)

**Author Response to Reviews of**

**A stochastic parameterization of ice sheet surface mass balance for the Stochastic Ice-Sheet and Sea-Level System Model (StISSM v1.0)**

L. Ultee, A. Robel, S. Castruccio
*Geoscientific Model Development*
* * *
RC: *Reviewer Comment*,     AR: *Author Response*,     ☐ Manuscript text

We thank the reviewer and editor for their comments. We respond to the remaining points below.

**1.  Reviewer 1**

**1.1.  Summary**

RC: *The revised version of the manuscript addresses all of my earlier concerns (except two) and has greatly improved. I congratulate the authors to their thorough work. I only have a few minor remaining points that I want to (re-)raise.*

AR: *We thank the reviewer for their renewed attention to our manuscript. We know that the manuscript and review response are long, and the reviewer undoubtedly has many demands on their time.*

**1.2.  Specific comments**

RC: *l93. "sum to annual time scales" reads like a repetition to l90. It could be made clearer that l90 is a summary statement of what is described in more detail below, like so: "We aggregate each SMB model output field for each outlet glacier catchment at an annual time scale. In order to achieve that, we overlay each field ..."*

AR: *Good point. Revised to "We aggregate each SMB model output field for each outlet glacier catchment at an annual time scale. **To achieve that,** we overlay each field..."*

RC: *l175 "Figure 3b shows example best fits for four model types and their BIC (see legend)." – I still find it very difficult to understand that the black line (AR0) in Figure 3b comes out as a good fit to the time series. If that is true, than Fig. 3b is clearly not a good illustration of that fitting result. Could you explain in the context of l175 why a model producing the time series without any variability is a good fit to the noisy time series in 3b? If reproducing the noise is not relevant for a good fit, what is?*

AR: *The reviewer is asking about fitting, which is in essence a question about what we are selecting for by optimizing the Bayesian Information Criterion. We address that in the Discussion, lines 323-326:*

> *Moreover, among low-order autoregressive models, white noise AR(0) models with a trend are preferred over higher-order models in most basins, for all seven process models tested (Figure A1). Low-order AR models could have a low BIC despite relatively greater error than higher-order models, as seen in Figure 3, because the BIC penalizes excess parameters (Equation 2).*

*We interpret that there is also a question about what the resulting generator will (re)produce, relative to the training series. We have added the following clarification to lines 179-181:*

> *We note that the models capturing only a trend in Figure 3b will still generate stochastic series with temporal variability; the distinction is that almost all of the temporal variability in the final generator will come from the spatial noise generation process described in the next section.*

**RC:** ***l295 "The same principle could be adapted for training data provided at even finer temporal resolution (i.e., weekly or daily)." I have remarked before that going to different time resolution may not be obvious because of other types of variability and difficulties to produce enough training data. ... I couldn't find a discussion item that specifically addresses problems related to going to a daily time scale. As it stands, I still find the statement above to easily made and not justified without proof that this is actually possible without additional problems.***

AR: *We have revised line 295 to be more general:*

> *The same principle could be adapted for training data provided at even finer temporal resolution, **though a large training data set may be needed to capture the relevant variability in sub-monthly SMB.***

*The point about choice of training dataset is further addressed in lines 391-397:*

> *The example presented here illustrates the possibility of inferring a downscaling function from process-model output. It would be possible to infer similar downscaling functions at different temporal or spatial resolutions, using reanalysis or reconstructed data, or computed over a different reference period. Ultimately, the choice of a reference period and the best spatial dataset to infer such a function depends on the user's intended application, and this selection may be non-trivial. Further, our simple downscaling does not capture changes in elevation dependence of SMB over time, for example due to changes in precipitation phase or local atmospheric lapse rate. Users seeking improved fine-scale performance may wish to implement more granular statistical downscaling methods (e.g. Noël et al., 2016).*